# Experimental evolution of yeast shows that public-goods upregulation can evolve despite challenges from exploitative non-producers

Richard J. Lindsay [1], Philippa J. Holder[1], Mark Hewlett[1] & Ivana Gudelj [1] ✉

Microbial secretions, such as metabolic enzymes, are often considered to be cooperative public goods as they are costly to produce but can be exploited by others. They create incentives for the evolution of non-producers, which can drive producer and population productivity declines. In response, producers can adjust production levels. Past studies suggest that while producers lower production to reduce costs and exploitation opportunities when under strong selection pressure from non-producers, they overproduce secretions when these pressures are weak. We challenge the universality of this trend with the production of a metabolic enzyme, invertase, by *Saccharomyces cerevisiae*, which catalyses sucrose hydrolysis into two hexose molecules. Contrary to past studies, overproducers evolve during evolutionary experiments even when under strong selection pressure from non-producers. Phenotypic and competition assays with a collection of synthetic strains - engineered to have modified metabolic attributes - identify two mechanisms for suppressing the benefits of invertase to those who exploit it. Invertase overproduction increases extracellular hexose concentrations that suppresses the metabolic efficiency of competitors, due to the rate-efficiency trade-off, and also enhances overproducers' hexose capture rate by inducing transporter expression. Thus, overproducers are maintained in the environment originally thought to not support public goods production.

Microbes secrete numerous products to modify the external environment for their benefit, including breaking down complex nutrients before importing the resulting metabolites[1–4]. These secreted products are frequently considered to be cooperative public goods because they are costly to produce, and their extracellular activity benefits neighbouring individuals. However, this presents an opportunity for non-producing individuals to gain a selective advantage by selfishly reaping the benefits from others without incurring the cost of production[2,5–12]. Non-producers can drive a decline in public-good producers and are often detrimental to the functioning of populations by hindering growth rates, biomass production and virulence[5,6,8–15]. In turn, producers can prevent being outcompeted by non-producers by gaining preferential access to any generated benefits[5,6,9,16]. This can be achieved by prudently regulating the expression of public goods to avoid wasteful production[17–21], by coupling public-good production with secondary private benefits[22,23], by punishing non-producers with toxins[14,24], or by de-repressing non-social traits that enhance their growth rate[25,26]. Understanding the evolutionary dynamics of public-

[1]Biosciences and Living Systems Institute, University of Exeter, Exeter, UK. ✉e-mail: I.Gudelj@exeter.ac.uk

good production is key to developing novel, sustainable management strategies that harness evolutionary principles for diverse purposes. These include treating infectious diseases[2,27], promoting healthy gut microbiomes[28], and utilising microbial consortia for bioremediation[29] and bioproduction[30].

Microbes can adjust public-good production levels in response to their environment, which can arise through either phenotypic plasticity[18,21] or genetic mutations[31,32]. If under strong selection pressure from non-producers (i.e., when non-producers can outcompete producers), such as when spatial structuring is low, producers can reduce the costs of cooperation, as theory predicts[33], by reducing public good production[17–20,27,31,34–36]. Conversely, producers can increase public-good production when they are not exposed to strong selection pressure from non-producers (i.e., when non-producers cannot outcompete producers), such as in spatially structured environments that usually promote the evolution of public-good production[5,15,27,32,37–42]. This upregulation provides benefits to the population, such as increasing access to resources[15,38,39] or suppressing interspecific competitors with antibiotics[32] or non-exploitable siderophores[43]. However, it can be costly to individual fitness when challenged by non-producers[32,39,40] because increased production incurs higher costs[33]. Thus, existing literature suggests that public-good upregulation will not evolve in environments where producers can be outcompeted by non-producers. To test this hypothesis, we considered public good production of invertase by *Saccharomyces cerevisiae*.

Diverse microbes secrete invertase to catalyse the hydrolysis of glycosidic bonds in carbohydrates, including sucrose[3,4]. Invertase is encoded by the *SUC2* gene in *S. cerevisiae*. SUC2-mediated extracellular sucrose hydrolysis has been widely used as a model system for investigating public-good production (e.g., refs. 8,9,38,44,45). Importantly, experiments with engineered SUC2 non-producers have demonstrated that they can invade populations of producers during competitions in shaken liquid media[9,46]. Thus, this system is well-suited for testing the above hypothesis because, through genetic and environmental modifications, we can control both the selection pressure on producers from non-producers, as well as the competitors' metabolic properties.

We found that after ~100 generations during batch culture serial transfer in well-shaken sucrose media, producers upregulated public-good production when cultured alone. This is consistent with previous findings in environments where selective pressures from non-producers are absent or weak[15,27,31,32,37–40,42,47]. We also considered well-shaken environments with both producers and non-producers present, where non-producers can outcompete producers over a single growth season. After equivalent serial transfers but in the presence of non-producers, producers also evolved to upregulate public-good production and prevented complete exclusion by non-producers. This was surprising because past studies have found that overproduction increases the relative fitness of non-producers and thus weakens the long-term stability of cooperation[17,32,39,40]. To identify the mechanisms driving this unexpected result, we conducted phenotyping and competition assays with a collection of genetically engineered, synthetic strains in the ancestral genetic background with different attributes for sucrose metabolism. We found that overproduction can bestow both direct and indirect benefits to producers that prevent them from being outcompeted by competitors. Namely, the increased hexose concentration from invertase overproduction increases the hexose capture rate of overproducers by inducing high-affinity transporter upregulation, and, in sufficiently high concentrations, it suppresses competitors' metabolic efficiency relative to overproducers through the convex geometry of the rate efficiency trade-off. This finding contrasts with past evolutionary experiments where producers downregulate production to reduce costs and exploitation by competitors[17–20,27,31,34–36].

## Results

### Generating a non-producer that exerts strong selection pressure on producers

*S. cerevisiae* expresses both an intracellular and secreted form of invertase from the same locus[48]. The latter is secreted into both the periplasm and the extracellular environment to externally hydrolyse sucrose into its constituent hexoses, glucose and fructose, that are imported in a quick and energetically cheap manner through hexose transporters[49]. To test our hypothesis that public-goods upregulation will not evolve in environments where producers can be outcompeted by non-producers, it was necessary to engineer a non-producer that exerts sufficiently strong selection pressure on producers for secreted products. However, previous studies have found that, in well-shaken sucrose media, SUC2 producers can coexist with *suc2*-deletion non-producers that have been generated in the commonly used S288C genetic background[9,13,45,46]. This is because the S288C non-producer is unable to effectively proliferate in sucrose media when alone and relies on exploiting SUC2 secreted by neighbouring producers, which facilitates coexistence with producers via negative frequency-dependent fitness[9,46]. Therefore, to develop a more competitive non-producer and overcome the S288C-based non-producer's sole reliance on secreted SUC2, we generated a *suc2*-deletion mutant in the wild-type CEN.PK2-1C genetic background (Supplementary Fig. 1). Unlike S288C, CEN.PK2-1C has an active *MAL* locus, so it can express both a high-affinity sucrose transporter (*AGT1*, alias of *MAL11*)[50] and an intracellular α-glucosidase (*MAL12*) that can hydrolyse sucrose[51]. Consequently, CEN.PK2-1C can slowly metabolise sucrose internally without *SUC2*, and the CEN.PK2-1C *suc2*-deletion mutant (henceforth "non-producer") successfully proliferates alone in sucrose media but at a slower rate than the wild-type (CEN.PK2-1C, henceforth "producer") (Supplementary Fig. 2a, b). Rather than coexisting[9,13,45,46], this newly generated non-producer outcompeted producers over a single 24 h batch culture season across a range of initial frequencies (Fig. 1a), which according to the well-established mathematical theory indicates the competitive exclusion of producers as the long term outcome[33,52]. Thus, this stronger competitor was used in order to exert strong selection pressure on producers.

### Producers adapt to prevent the loss of cooperative sucrose metabolism

To test the evolutionary responses of producers, we conducted multiple-season serial transfer evolutionary experiments of either producer alone or in mixed populations with non-producers over 10 seasons (~100 generations – $n = 3$). Mixed populations were initiated with ~10% producers to give them a lower probability of acquiring adaptive mutations in non-social traits compared to the numerically dominant non-producers[25,26,53]. An example of such a non-social trait in our scenario is sucrose uptake via *AGT1*, as discussed above. Although the frequency of producers initially declined significantly in mixed populations (from ~10 to ~6%), the competitors' relative fitness subsequently equalised over the course of the multiple-season experiment, allowing producers to persist (Fig. 1b). This is despite ancestral non-producers having a fitness advantage over ancestral producers when producers were at equivalent low frequencies during single-season competitions (Fig. 1a). This suggests that coexistence is not simply caused by negative frequency-dependent fitness between the ancestral strains, as is often found between competitors for secreted products[9,33], but rather that the producers have increased their relative fitness in this environment.

To identify potential fitness gains, we isolated six clones of the evolved producer strain, three (c1–c3) from mixed populations (shown in Fig. 1b) and three (a1–a3) from equivalently evolved populations but with producers alone (one clone per line). The fitness of these six evolved producers was compared with the ancestral producer (henceforth "wt") when each was in pairwise competition against the

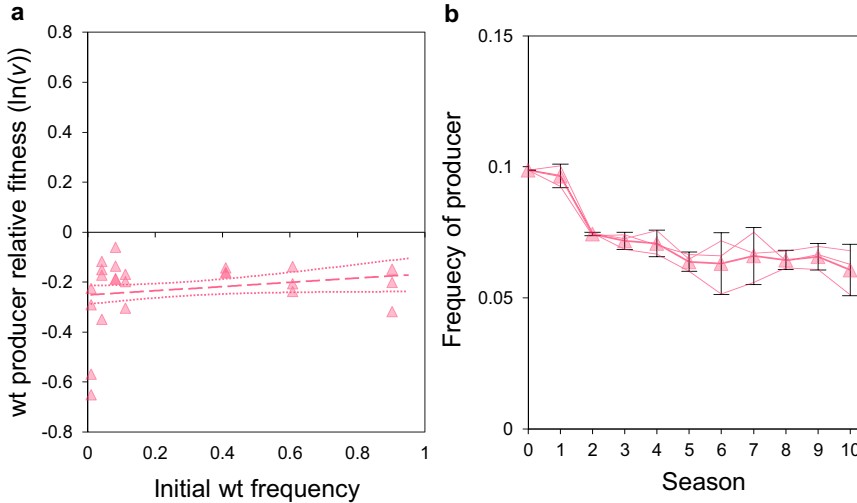

**Fig. 1 | Competitions between wt SUC2 producers and non-producers. a** Single season (24 h) competitions were conducted between producers and non-producers. Producers were less fit than non-producers (two-sided linear regression model (± S.E.): intercept: − 0.252 ± 0.037, $p = 8.65 \times 10^{-7}$; frequency dependence was not significant: $F_{(1,22)} = 0.885$, $p = 0.357$). Points show all replicates. Equal fitness = 0. Initial wt frequencies = 0.009, 0.041, 0.082 ($n = 4$), 0.111, 0.409, 0.608, 0.902 ($n = 3$). **b** Longer-term evolutionary competition experiments between producers and non-producers over approximately 100 generations. Populations were diluted

600x and transferred to fresh media between each season. The frequency of producers initially declined (two-sided linear regression model: significant negative relationship for season 0–10 ($p = 9.33 \times 10^{-8}$), season 1–10 ($p = 2.45 \times 10^{-5}$), season 2–10 ($p = 3.52 \times 10^{-3}$), and season 3–10 ($p = 4.26 \times 10^{-2}$)) before plateauing over the course of the experiment (two-sided linear regression model: non-significant from season 4–10 ($p = 0.213$)). Mean ± 95% C.I., $n = 3$ populations. Narrow lines track individual replicates. All data are provided in the Source Data File.

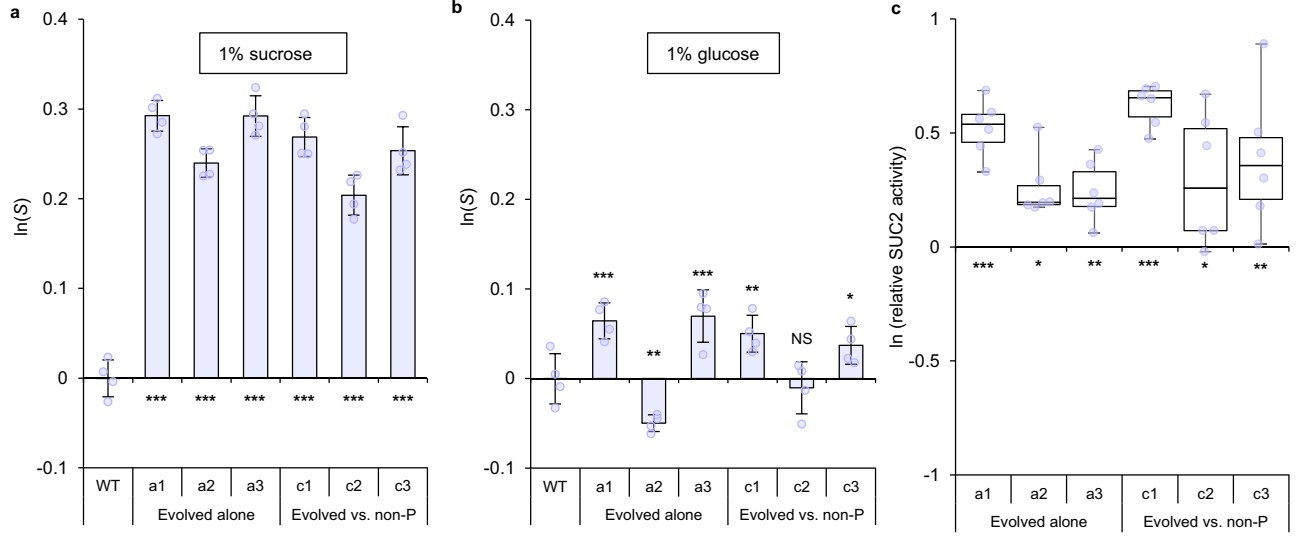

**Fig. 2 | Evolved producers had increased fitness in sucrose and SUC2 activity.** **a** Single season (24 h) pairwise competitions were conducted with WT or evolved producers against non-producers (initial frequency = 0.5). **a** All evolved producers tested were fitter than the WT in sucrose (GLM (two-sided): a1–a3, c1–c3: $p = 9.83 \times 10^{-15}$, $4.88 \times 10^{-13}$, $9.98 \times 10^{-15}$, $5.25 \times 10^{-14}$, $1.09 \times 10^{-11}$, $1.65 \times 10^{-13}$ – see Supplementary Fig. 3a for relative fitness vs. non-producers). **b** All clones lost or had a diminished competitive advantage over the WT in glucose (GLM (two-sided): glucose c.f. sucrose: $F_{(13,42)} = 125.7$, $p < 2.2 \times 10^{-16}$, Adj. R² = 0.967; a1–a3, c1–c3 c.f. 0 in glucose: $p = 9.70 \times 10^{-4}$, $7.87 \times 10^{-3}$, $4.63 \times 10^{-4}$. $7.22 \times 10^{-3}$, 0.559, $3.84 \times 10^{-2}$; glucose < sucrose (Tukey's HSD): a1–a3, c1–c3: $p = 1.16 \times 10^{-12}$, $1.06 \times 10^{-12}$, $1.07 \times 10^{-12}$, $1.07 \times 10^{-12}$, $1.08 \times 10^{-12}$, $1.07 \times 10^{-12}$). **c** Measured invertase activity was

normalised against the WT (Supplementary Fig. 3b). All evolved producers had increased invertase activity compared to the WT (GLM (two-sided): a1-3, c1-3: $p = 1.01 \times 10^{-5}$, $3.65 \times 10^{-2}$, $7.47 \times 10^{-3}$, $8.45 \times 10^{-5}$, $1.99 \times 10^{-2}$, $4.32 \times 10^{-3}$), but there was no significant influence of whether clones had evolved alone or in the presence of non-producers (GLM (two-sided): $p = 0.385$). **a–c** Asterisks indicate significant differences from the ancestral WT ( = 0): ***$p < 0.001$, **$p < 0.01$, *$p < 0.05$, NS $p > 0.05$; full analysis shown in Supplementary Data 1. Points show all replicates. **a, b** Bars show mean ± 95% C.I., $n = 4$, $S$ = normalised relative fitness. **c** Box-plots show 25, 50, 75th percentiles, whiskers show min/max, $n = 6$. All data are provided in the Source Data File.

ancestral non-producer (henceforth "non-producer"). All six evolved producers were fitter than the wt when competing in the same resource conditions that they had evolved in (1% sucrose) (i.e., ln($S$) > 0, Fig. 2a, where $S$ is a normalised relative fitness defined in Methods). These evolved producers also overcame the ancestral wt's

fitness disadvantage when competing against the ancestral non-producer (Supplementary Fig. 3a).

We next wanted to understand the nature of this fitness gain and, in particular, whether the evolved producers had adapted directly to the "public" (i.e., extracellular) nature of sucrose metabolism or to

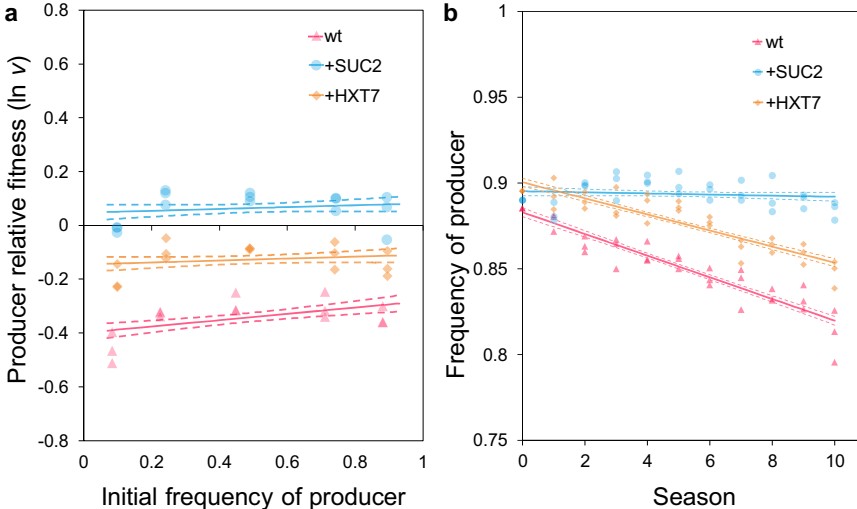

**Fig. 3 | The competitive ability of wt and engineered overproducers against non-producers.** Competition experiments in 1% sucrose SC media were conducted between either the ancestral producer (wt), or an otherwise isogenic strain engineered to overexpress *SUC2* (+SUC2) or *HXT7* (+HXT7), against the ancestral non-producer. **a** In single season (24 h) competitions, the ancestral producer was less fit than both +HXT7 (GLM (two-sided): $p = 4.99 \times 10^{-7}$) and +SUC2 ($p = 5.30 \times 10^{-13}$), while +SUC2 was fitter than +HXT7 ($p = 6.38 \times 10^{-5}$). WT ($p = 2.32 \times 10^{-16}$) and +HXT7 ($p = 2.47 \times 10^{-5}$) were less fit than the non-producer, yet +SUC2 overcame this fitness deficit (not significantly different from 0: $p = 0.129$). Points show all replicates, $n = 3$ for each frequency/competition combination. See Supplementary Fig. 5 for the control experiment on glucose, equivalent competition vs. wt producer, and full statistical analysis. **b** Multiple season competitions (10 seasons with 600x dilution between seasons ≈ 100 generations). Frequency was significantly associated with genotype and season (GLM (two-sided): $F_{(5,93)} = 169.9$, $p < 2.2 \times 10^{-16}$, Adj. $R^2 = 0.896$). See main text for statistical analysis of genotype comparisons. Points show all replicates, $n = 3$ for each time-point/competition combination. **a**, **b** Lines show linear models (± S.E.) for individual producer genotype competitions. All data are provided in the Source Data File.

other indirect, "private" (i.e., intracellular) features of the environment during the evolutionary experiment, e.g., enhanced nutrient uptake[26,54]. To achieve this, the wt and evolved producers were again each competed in a pairwise fashion against the non-producer, but this time the sucrose was replaced by glucose (1%), which all competitors directly uptake via hexose transporters[55]. In glucose, although 4 evolved producers (2 evolved with non-producers and 2 evolved alone) maintained a competitive advantage over the wt, the magnitude of this advantage was significantly reduced compared to the equivalent measure in the "public" sucrose environment (Fig. 2b and Supplementary Data 1). Moreover, one evolved producer (c2) lost its competitive advantage in glucose, while another was less fit than the wt (a2) (Fig. 2b and Supplementary Data 1). These results suggest that the competitive advantage that the evolved producers gained over the wt predominantly relates to how they metabolise sucrose.

### Evolved producers have increased fitness and invertase secretion
Secreted invertase activity levels of the evolved producers were compared to the wt using a colourimetric enzyme assay. All 6 evolved producers had elevated secreted invertase activity compared to the wt. These increases were independent of whether producers had evolved in the presence or absence of non-producers (Fig. 2c and Supplementary Fig. 3b).

To test whether the enhanced competitiveness of the evolved producers resulted from elevated invertase activity, a strain (namely +SUC2) was generated in the wt genetic background that possessed an additional copy of *SUC2* under the regulation of the strong and constitutive *GPD* promoter. +SUC2 had upregulated secreted invertase activity (Supplementary Fig. 4) and had an increased growth rate in 1% sucrose media compared to the wt (Supplementary Fig. 2). When competing against the non-producer, +SUC2 was a superior competitor than the wt (Fig. 3a). We tested whether this competitive difference was the result of sucrose metabolism, rather than, for example, off-target effects of the genetic modification, by repeating the competitions with glucose instead of sucrose. The significant competitive

differences that were detected in sucrose media (Fig. 3a) were lost in glucose media (Supplementary Fig. 5a), confirming that the competitive differences between strains are caused by differing sucrose metabolism. Given that overproducers also evolved in the absence of non-producers (Fig. 2c), +SUC2 was also competed pairwise against the wt producer in 1% sucrose media, with +SUC2 found to be the stronger competitor (Supplementary Fig. 5b). Thus, as anticipated from the evolved overproducers (Fig. 2a, c), SUC2 overproduction bestows fitness benefits when overproducers compete against non-producers and/or producers that produce comparatively less invertase than overproducers. Such producers can be considered as "cheats" in this context[56].

The competitiveness of +SUC2 and its ability to resist invasion by non-producers was also tested by conducting multiple-season competition experiments, with non-producers at an initial frequency of 10% (Fig. 3b). This frequency was used to promote non-producer relative fitness due to negative frequency-dependent selection[33]. In agreement with the short-term competitions, +SUC2 was resistant to invasion (linear regression model for +SUC2: β (± S.E.) = − 0.0003 (±0.0004), $F_{(1,31)} = 0.553$, $p = 0.463$) and a better competitor than the wt (GLM: genotype:season interaction term; +SUC2 > wt: $p < 2 \times 10^{-16}$). This supports the hypothesis that upregulating *SUC2* expression has increased the competitive ability of the evolved producers.

### Why does public-good overproduction promote producers?
By considering *S. cerevisiae* sucrose metabolism, we hypothesised that *SUC2* overproduction might enhance producer competitiveness in three non-mutually exclusive ways (Fig. 4a, b): 1) It concurrently increases overproducers' private sucrose metabolism (Fig. 4b1). 2) It suppresses the benefits of exploitation by reducing competitors' metabolic efficiency (Fig. 4b2). 3) It concurrently increases overproducers' hexose capture efficiency (Fig. 4b3). Each was tested by conducting competition experiments with either altered sucrose concentrations in the media, or with a collection of synthetic strains with altered metabolic attributes (summarised in Table 1, more details in methods and Supplementary Data 3). This was achieved, as

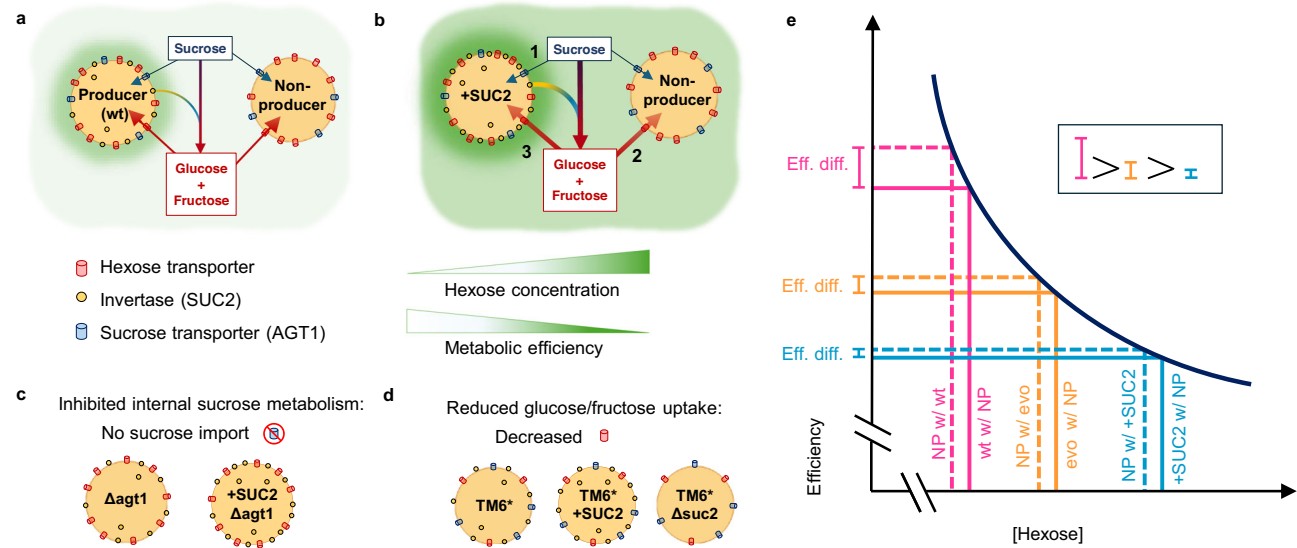

**Fig. 4 | Schematics of the experimental system (a, b), and experimental (c, d) and theoretical (e) approaches used to identify mechanisms promoting overproduction. a** Wild-type (wt) producers metabolise sucrose by either secreting invertase to hydrolyse sucrose and import the products (glucose + fructose) through hexose transporters or by directly importing sucrose and hydrolysing it with either internal invertase (SUC2) or α-glucosidase (MAL12 - not shown). Non-producers can also metabolise sucrose internally with α-glucosidase, or exploit the glucose + fructose generated by producers. **b** + SUC2 increases sucrose hydrolysis and hexose concentrations, which reduces metabolic efficiency. + SUC2 could hypothetically increase fitness in 3 ways (thicker arrows indicate higher rates): **1** Increase internal metabolism by increasing internal invertase production. **2** Competition suppression by imposing inefficient metabolism on exploiters ("Mechanism 1"). **3** Increase glucose + fructose capture efficiency ("Mechanism 2"). **c, b1** was tested by deletion of the sucrose transporter *AGT1*, thus preventing sucrose import.

**d, b3** was tested using strains with reduced hexose uptake capacity. **e** Schematic illustrating the theoretical metabolic efficiencies of competitors, which is the basis of (**b2**): Efficiency decreases as [hexose] (and thus hexose uptake) increases in a convex manner (ref. 62; Fig. 6b). Non-producers (NP) experience lower [hexose] than the producers (P) they exploit because P uptake a portion of generated hexose before it becomes communal[9]. Therefore, hexose uptake by NP when cocultured with (w/) P - either wt (dashed pink), evolved (evo) (dashed orange), or + SUC2 (dashed blue) - are situated to the left of the solid lines that represent hexose uptake by P – either wt (solid pink), evo (solid orange), or + SUC2 (solid blue), respectively, when they are cocultured with NP. While both P + NP strains suffer reduced efficiency in populations containing overproducers (+ SUC2 (blue) or evo P (orange)) c.f. wt (pink), the convex geometry of the trade-off means the efficiency difference (Eff. diff.) between P and NP is smaller in overproducer populations. **b2** was tested with competitions in 0.05% Sucrose.

## Table 1 | Summary of subset of strains generated in this study

| Strain name | Metabolic genetic modifications | Characteristics | Motivation for generating strain(s) |
|---|---|---|---|
| wt | n/a | *MAL*-constitutive SUC2 producer | Wt producer |
| Non-producer | *suc2Δ::kanMX* in wt genetic background | *MAL*-constitutive SUC2 non-producer | To impose selection for public goods |
| + SUC2 | Constitutively expresses an additional copy of *SUC2* | SUC2 overproducer in wt genetic background | To test whether overproducing SUC2 bestows direct benefits |
| + HXT7 | Constitutively expresses an additional copy of *HXT7* | HXT7 overproducer in wt genetic background | To test the influence of increased, high-affinity hexose transporter expression on producer fitness |
| Δagt1 | *agt1Δ::kanMX* | SUC2 producer without sucrose importer | To test whether SUC2 overproducers gain fitness advantage from upregulating "private" internal sucrose metabolism |
| + SUC2Δagt1 | *agt1Δ::kanMX* | SUC2 overproducer without sucrose importer | |
| Δsuc2Δagt1 | - *suc2Δ::kanMX*<br>- *agt1Δ::TRP1* | SUC2 non-producer without sucrose importer | Verification that Δagt1 & + SUC2Δagt1 metabolise sucrose externally since this strain doesn't grow in sucrose media. |
| TM6* | The previously generated strain only has a single synthetic hexose transporter gene | SUC2 producer with reduced hexose uptake capacity | To test whether SUC2 overproducers gain a competitive advantage from enhanced sucrose hydrolysis product capture. |
| TM6*Δsuc2 | *suc2Δ::kanMX* | SUC2 non-producer with reduced hexose uptake capacity | |
| TM6* + SUC2 | Constitutively expresses an additional copy of *SUC2* | SUC2 overproducer with reduced hexose uptake capacity | |

For a full set and more details, see Supplementary Data 3 and the main text. Italic formatting is used for gene names.

explained below (see also Fig. 4a–d), by altering a secondary feature of sucrose metabolism in both a strain with wt SUC2 production and one with + SUC2, and then comparing each producer's competitiveness while in pairwise competition against non-producers.

In addition to the secreted form of SUC2, *S. cerevisiae* produces a form of SUC2 that lacks a secretion signal peptide and so remains intracellular and hydrolyses sucrose that is imported in a slow and energetically expensive manner via AGT1[46,48,50]. Thus, we first ruled out

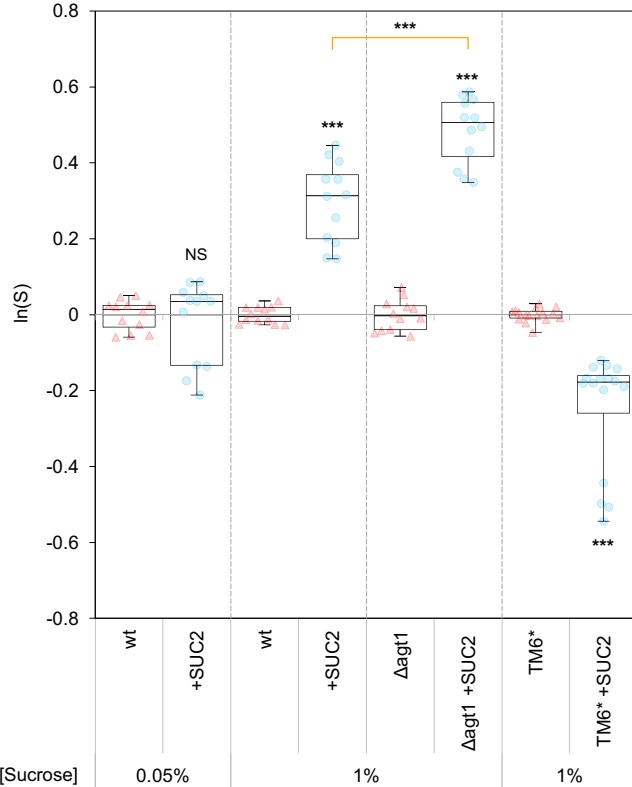

**Fig. 5 | The normalised relative fitness of synthetic producer strains with different sucrose metabolism attributes.** Fitnesses (as normalised relative fitness ($S$) – see "Methods") of different producer genotypes were measured in competition with otherwise equivalent non-producer ($\Delta suc2$) strains. Points show all replicates, box plots show 25, 50, 75th percentiles, whiskers show min/max, $n = 12$ except for TM6* strains when $n = 16$. Generalised Linear Model (two-sided) with post-hoc Tukey (HSD method): NS, not significant ($p > 0.05$), \*\*\*$p < 0.001$ significant differences from 0, or between genotypes indicated with orange line. Exact $p$-values: wt vs. + SUC2 (0.05, 1%) = 0.999, 1.45 x 10⁻¹³; $\Delta agt1$ vs. $\Delta agt1$ + SUC2 = < 10⁻¹⁵; TM6* vs. TM6* + SUC2 = 3.75 x 10⁻⁸; + SUC2 vs. $\Delta agt1$ + SUC2 = 2.39 x 10⁻¹⁰. The full analysis is shown in Supplementary Data 2. All data are provided in the Source Data File.

that the increased competitiveness of SUC2 overproduction arises from increasing the rate of internal, "private" sucrose metabolism (Fig. 4b1). This was achieved by deleting *AGT1* from both the wt and + SUC2 strains to prevent sucrose uptake, generating strains $\Delta agt1$ and + SUC2$\Delta agt1$, respectively (Fig. 4c and Supplementary Fig. 6). AGT1 has previously been shown to increase the growth rate of *S. cerevisiae* cells growing on sucrose[50], and both $\Delta agt1$ and + SUC2$\Delta agt1$ were found to have lower growth rates than their respective progenitors (Supplementary Fig. 7). Moreover, the *AGT1/SUC2* double mutant ($\Delta suc2\Delta agt1$ – Supplementary Fig. 7d) did not grow in sucrose (Supplementary Fig. 7b). Therefore, we demonstrated that AGT1 enabled sucrose uptake, and internal sucrose metabolism was effectively blocked in $\Delta agt1$ and + SUC2$\Delta agt1$.

We then compared the competitiveness between the wt and + SUC2, and between $\Delta agt1$ and + SUC2$\Delta agt1$, all while in pairwise competition against the non-producer in 1% sucrose media. We found that inhibiting internal sucrose metabolism in the *agt1*-deletion strains did not prevent the invertase overproducer (+ SUC2$\Delta agt1$) having a selective advantage over its equivalent wt producer ($\Delta agt1$) (ln($S$) > 0, Fig. 5). In addition, + SUC2$\Delta agt1$ had higher normalised relative fitness than the overproducer with intact internal sucrose metabolism (+ SUC2) (Fig. 5 and Supplementary Data 2). Moreover, when AGT1 was deleted, the wt producer suffered a larger reduction in axenic growth

rate (wt versus $\Delta agt1$) than the overproducers (+ SUC2 versus + SUC2$\Delta agt1$) (Supplementary Fig. 7). Collectively, these results suggest that the wt producer has a higher reliance on internal sucrose metabolism than + SUC2 and that the enhanced competitiveness of the SUC2 overproducers is not caused by simultaneously increasing internal sucrose metabolism. Therefore, we next focused on how SUC2 overproduction could enhance fitness via external sucrose metabolism and identified two mechanisms.

### Mechanism 1: SUC2 overproduction suppresses the benefits of exploitation

We reasoned that SUC2 overproducers might gain a competitive advantage by repressing competitors to reduce the benefits they obtain[57,58]. Like many microbes, *S. cerevisiae* experiences a rate-efficiency trade-off when metabolising sugars whereby increased resource uptake and growth rate coincide with reduced metabolic efficiency[59–63]. The geometry of this trade-off is convex, whereby as resource uptake increases, the reduction in efficiency plateaus[61,62] (see Fig. 4e for an illustration).

In particular, during extracellular SUC2-mediated sucrose metabolism, *S. cerevisiae* producers capture a portion of the generated hexose before the remaining diffuses into the environment to be globally available[9] (Fig. 4a). This extra hexose capture means that producers metabolise resources with a lower efficiency than other cells which only uptake hexose generated by others[44,64]. When SUC2 overproducers increase the rate of extracellular sucrose hydrolysis, we reasoned that although all cells experience a reduction in metabolic efficiency, the convex geometry (rather than linear) of the rate-efficiency trade-off means that the efficiency difference (Eff. diff.) between non-producers and their competitors is larger when non-producers are exploiting the wt compared to when they exploit SUC2 overproducers (Fig. 4e, pink Eff. Diff. > blue Eff. Diff). Indeed, firstly by examining the final densities of axenic populations of our strains with different metabolic rates, a between-strain rate-efficiency trade-off was evident whereby faster growing strains had lower metabolic efficiency, thus causing a reduced population yield (carrying capacity per unit of sugar) (Supplementary Fig. 2c). Secondly, the environmental hexose concentration created from sucrose hydrolysis was increased by + SUC2 compared to the wt producer over the course of a batch culture growth season (Fig. 6a). Thus, competitors co-cultured with +SUC2 would experience increased hexose concentrations, which leads to reduced growth efficiency[62], as demonstrated by a decline in wt population yields as glucose concentration increases (Fig. 6b). We then tested the ability of either the wt or + SUC2 to suppress the efficiency of the non-producers by measuring the carrying capacity of mixed populations of non-producers with either the wt or + SUC2, initiated at different frequencies on 1% sucrose. Consistent with our hypothesis, we found that while both the wt and + SUC2 (when at sufficiently high frequencies in the population) suppress the high efficiency of the slower-growing non-producers, + SUC2 suppressed the efficiency of the mixed population to a much larger extent than the wt (Fig. 6c).

To test whether this suppression drives the competitive benefit of SUC2 overproduction, we again compared the competitive ability of the wt and + SUC2 when each producer genotype was in pairwise competition with non-producers, but this time with a lower sucrose concentration (0.05%) where the rate-efficiency trade-off is weak or absent (Supplementary Fig. 8)[62]. In line with the competition suppression hypothesis (Mechanism 1), + SUC2 lost its selective advantage over the wt in pairwise competition with non-producers when the trade-off was removed (Fig. 5 – 0.05% sucrose vs. 1% sucrose). However, + SUC2 had equivalent fitness to the wt (i.e., ln($S$) ≈ 0), which was unexpected for two reasons. First, since overproduction is anticipated to incur increased metabolic costs[17], + SUC2 would be expected to have lower fitness than the wt (ln($S$) < 0) when its superior ability to

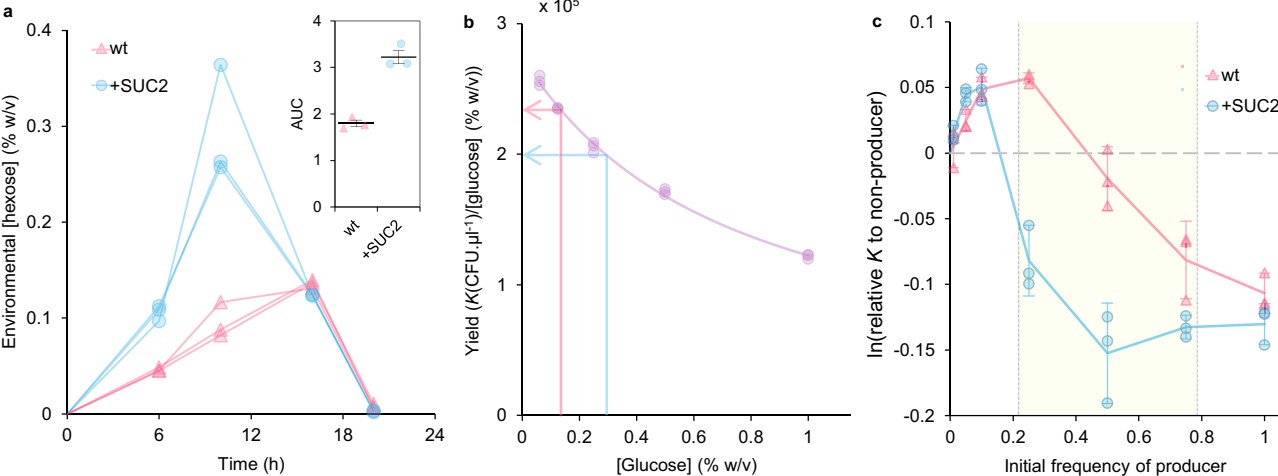

**Fig. 6 | Overproducers suppress competitors by reducing their metabolic efficiency. a** Environmental hexose concentration during batch culture season in 1% sucrose. Hexose concentration in supernatant: + SUC2 > wt (Area Under Curve (inset): Two-sample, two-sided $t$ test: $t = 9.00$, $p = 8.45 \times 10^{-4}$), $n = 3$. **b** Growth yield of wt decreases with increasing resource concentration in a non-linear manner, which also applies to + SUC2 and non-producers because they consume glucose equivalently. The purple line is fitted from Eq. (1) of ref. 61. Arrows indicate yield of mean maximum [hexose] from supernatant of wt (red, 16 h: 0.136 ( ± 0.005) %) or + SUC2 (blue, 10 h: 0.295 ( ± 0.068) %) from (**a**) to demonstrate the estimated growth efficiencies of populations with either wt or + SUC2 producers. **c** Population carrying capacity ($K$ - density after 36 h) of mixed populations were suppressed to a greater extent by + SUC2 than wt, (GLM (two-sided): $F_{(13,28)} = 66.19$, $p < 2.2 \times 10^{-16}$, Adj. $R^2 = 0.95$; post-hoc Tukey's HSD: Exact $p$-values for significant between-genotype differences in shaded region: $0.25 = 4.43 \times 10^{-9}$, $0.5 = 1.15 \times 10^{-8}$, $0.75 = 4.00 \times 10^{-2}$). Points show all replicates ($n = 3$), lines follow mean ± 95% C.I. The dashed horizontal line shows the mean yield of pure non-producer populations. All data are provided in the Source Data File.

suppress non-producers is removed. Second, increased sucrose hydrolysis, as shown by the higher growth rate of + SUC2 compared to the wt in 0.05% sucrose (Supplementary Fig. 8b), should offer increased exploitation opportunities for non-producers, leading to + SUC2 fitness being lower than the wt (ln(S) < 0). Therefore, in summary, these results suggest that while competition suppression promotes the success of overproducers, it is not the lone mechanism driving the advantage because + SUC2 does not suffer an expected disadvantage compared to the wt when its influence is inhibited.

## Mechanism 2: SUC2 overproduction increases the portion of hexose captured by producers

Since sucrose hydrolysis occurs predominantly at the cell surface, SUC2-secreting cells have been estimated to capture a small portion of the generated hexose before it is communally available[9]. Although it has been suggested that this portion is independent of SUC2 activity levels[9], competitors for limited resources might enhance their fitness by increasing resource uptake[38,54], and the high-affinity hexose transporter HXT7 has previously been shown to be under positive selection in both glucose-[54] and sucrose-limited[38] environments. Thus, to explore the potential role of increased product capture on the producer's competitiveness, an additional copy of *HXT7*, also regulated by the *GPD* promoter, was introduced into the wt (creating strain + HXT7). When competing against the non-producer, + HXT7 was a superior competitor than the wt, as expected from an enhanced capacity to capture hexose (Fig. 3a). In multiple-season competition with non-producers at an initial frequency of 10% (Fig. 3b) we found that although non-producers could increase in frequency against +HXT7 (linear model for + HXT7: β (±S.E.) = − 0.047 ( ± 0.0004), $F_{(1,31)} = 135.1$, $p < 7.83 \times 10^{-13}$), they did so at a slower rate than when competing with the wt (GLM: genotype:season interaction term, + HXT7 > wt: $p < 7.96 \times 10^{-3}$). These results support the premise that producers can enhance their competitiveness by enhancing product capture.

We next tested whether SUC2 overproduction does increase the proportion of hexose that is captured by cells before it becomes communally available (the hexose capture efficiency ($\varepsilon$), fructose is considered equivalent to glucose). This was calculated using an equivalent approach to that used previously[9], which estimates the proportion of hexose molecules that are imported from extracellular sucrose hydrolysis at dilute cell densities. We found that + SUC2 had greater hexose capture efficiency than the wt (Fig. 7a and Supplementary Fig. 9).

This raised the question of how SUC2 overproduction increases hexose capture. *S. cerevisiae* has a large family of hexose transporters with different substrate affinities, that are dynamically regulated in response to glucose concentration. The hexose transporter, *HXT2*, was identified as a candidate for enabling + SUC2 to increase hexose capture. *HXT2* has a high affinity for glucose and is upregulated in response to increasing glucose concentrations within the low concentration ranges that were estimated to be surrounding cells during sucrose metabolism (Supplementary Fig. 9)[55,65]. Moreover, *HXT2* was the predominant high-affinity hexose transporter with increased expression in cells previously evolved under sucrose selection[38]. To measure how *HXT2* expression varied between the wt and + SUC2, we introduced constructs containing *HXT2*-promoter-regulated *EGFP*. We found, as previously[65], that *HXT2* expression is initially induced by increasing glucose concentrations but repressed by higher glucose concentrations. This expression pattern was similar in both the wt and + SUC2 (Supplementary Fig. 10a–c). However, when tested in 1% sucrose, + SUC2 had significantly increased *HXT2* expression compared to the wt (Fig. 6b and Supplementary Fig. 10d), and this qualitative difference was consistent when starter cultures were supplemented with either 1% galactose or 4% glucose, both of which are conditions that repress *HXT2* expression[65]. *HXT2* expression was also higher for + SUC2 than the wt in 0.05% sucrose (Supplementary Fig. 10e–h), suggesting that + SUC2 also benefits from additional hexose capture in low sucrose conditions. Therefore, the fitness advantage that + SUC2 has in 1% sucrose was not lost in 0.05% due to a loss of extra hexose-capture ability. Rather, this loss of advantage stems from the loss of ability to suppress competitors through the rate-efficiency trade-off (Mechanism 1) because the trade-off is weakened in low-resource environments (Supplementary Fig. 8)[62]. We attribute increased *HXT2* upregulation in + SUC2 over the wt to the increased hexose concentrations generated from SUC2

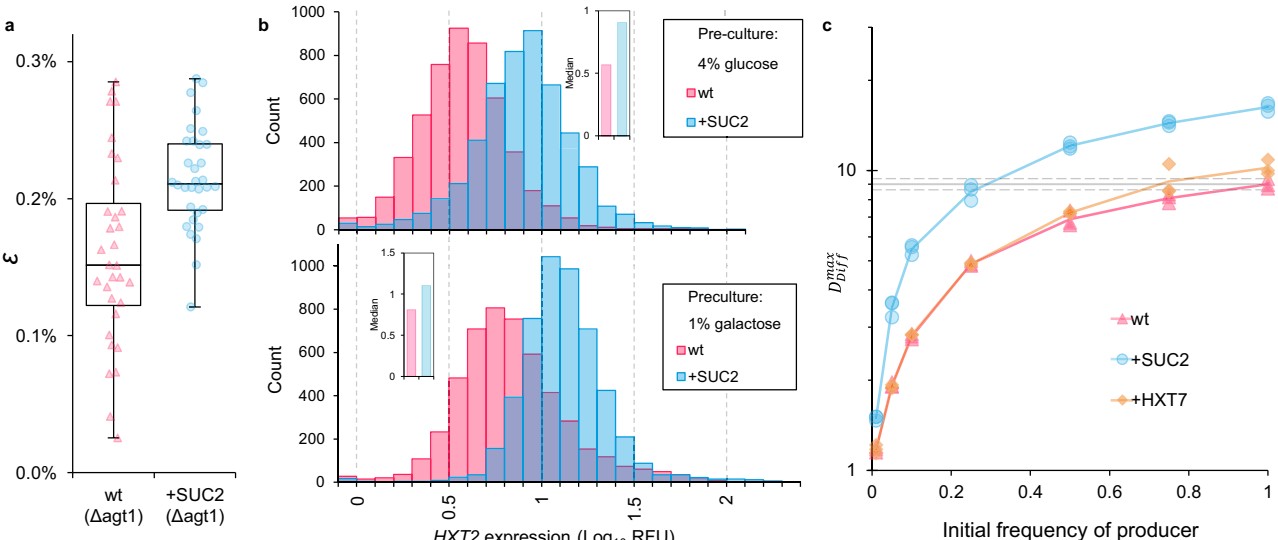

**Fig. 7 | Overproducers increase product capture via transporter upregulation, and enhance population growth. a** SUC2 overexpression increases glucose capture efficiency ($\varepsilon$) (Welch's two-sided $t$ test: $p = 2.86 \times 10^{-4}$, $t = 3.91$, $n = 32$ populations), min/max, 25/50/75th percentiles. **b** *HXT2* expression is higher in +SUC2 than wt during growth on 1% sucrose, when precultured in 4% glucose (top) or 1% galactose (bottom), as measured by *HXT2*-promoter-regulated *EGFP* (see Supplementary Fig. 10 for [glucose] expression profiles, experimental repeats, and expression in 0.05% sucrose). Bars show expression distribution of 5000 cells per strain. Insets show median values of 5000 events. **c** Maximum difference in population density ($D_{Diff}^{max}$, plotted on Log$_{10}$ scale) over batch culture in 1% sucrose

of populations initiated with different frequencies of each producer genotype, relative to pure non-producer populations (=1) (See Supplementary Fig. 13). Lines follow the mean for each producer genotype, points show all replicates, $n = 3$. The horizontal line shows a mean ± 95% C.I. for pure wt producers. $D_{Diff}^{max}$ was significantly associated with an initial frequency of producer and producer genotype (GLM (two-sided): $F_{(5,57)} = 212.4$, $p < 2.2 \times 10^{-16}$, Adj. $R^2 = 0.945$), genotype:frequency interaction term = wt vs. + HXT7: $p = 0.133$; wt vs + SUC2: $p = 1.65 \times 10^{-9}$. Post-hoc Tukey's (HSD): 100% wt < 100% + HXT7: $p = 3.11 \times 10^{-2}$; 50% + SUC2 > 100% wt: $p = 9.06 \times 10^{-11}$; 75% and 100% + SUC2 > 100% wt: $p = 1.06 \times 10^{-12}$. All data are provided in the Source Data File.

overproduction, which are within the low hexose concentrations that induce *HXT2* expression (Supplementary Figs. 9a, e, 10a–c). This suggests that +SUC2 can capture a larger proportion of hexose from external sucrose hydrolysis by simultaneously upregulating high-affinity hexose transport along with SUC2.

To test whether increased glucose capture efficiency during SUC2 overproduction enhances competitiveness, we generated another SUC2 overproducing strain, using the same construct as before. This time, however, the construct was introduced into a mutant strain of the wt (TM6*). TM6* possesses only a single hexose transporter gene, so it has a substantially reduced hexose uptake capacity[66] and is less able to capture hexose released from sucrose hydrolysis before it becomes communally available[44] (Fig. 4d). Like + SUC2, this new strain (namely, TM6*+SUC2) had increased invertase activity compared to its progenitor (TM6*) (Supplementary Fig. 11a), which increased its growth rate on 1% sucrose media (Supplementary Fig. 11c, d). We then compared the competitiveness of this overproducer (TM6*+SUC2) with the wt (TM6*) when in pairwise competition against an otherwise isogenic *suc2*-deletion strain (TM6*Δsuc2) that has impaired invertase activity (Supplementary Fig. 11b). In support of Mechanism 2, we found that this overproducing strain with impeded hexose capture ability was less competitive than its progenitor (Fig. 5 - TM6* c.f. TM6*+SUC2).

Since increasing sucrose concentrations can promote non-producers[44], we then tested the sensitivity of the fitness gain of +SUC2 to increases in hexose creation by increasing sucrose concentrations from 1 to 4% (Supplementary Fig. 4). Although +SUC2 maintained its competitive advantage over wt in 4% sucrose, its magnitude was reduced (Supplementary Fig. 12). We attribute this to non-producers capturing a larger proportion of the hexose generated because producer axenic growth rates were equivalent (Supplementary Fig. 7). This provides further evidence that + SUC2 fitness gains are sensitive to the proportion of hexose they capture.

## The impact of SUC2 overproduction on population growth

The slow growth associated with non-production of public goods can threaten the stability of microbial populations in transient environments[46], analogous to the "cheater-load"[12] (i.e., the decrease in population growth as a result of a non-producing subpopulation). To test the impact of different engineered producer genotypes on overall population growth when competing with non-producers, we examined the growth of populations of the wt, + SUC2 or + HXT7 that were initiated with different frequencies of non-producers. We calculated the resilience of population growth to the presence of non-producers ($D_{Diff}^{max}$, Supplementary Fig. 13) and found that although growth increases with increasing producer frequency in a non-linear manner, it has not completely saturated (Fig. 7c). Moreover, increased hexose capture by overexpressing *HXT7* does enable increased population growth to a small extent (Fig. 7c), but does not provide a significant increase in resilience to population growth against the threat of non-producers[46], i.e., $D_{Diff}^{max}$ values were not significantly different whether producers were wt or + HXT7 (GLM: genotype:frequency interaction term (wt vs + HXT7), $p = 0.133$). On the other hand, increased invertase production provides substantial resilience against non-producers (Fig. 7c, GLM: genotype:frequency interaction term (wt vs + SUC2), $p = 1.65 \times 10^{-9}$), with growth matching pure wt populations when + SUC2 is at a frequency of 25% (post-hoc Tukey's: 100% wt vs. 25% + SUC2, $p = 0.977$). Moreover, when + SUC2 was in populations at frequencies > 25%, they outgrew pure wt populations.

## Discussion

Microbes can adjust public goods production levels in response to their environment. Current theoretical and empirical literature supports the hypothesis that public goods overproduction will not evolve and persist in environments where non-producers can outcompete producers. Indeed, increasing levels of secreted products are predicted to be especially vulnerable to invasion by non-producers[33],

which is supported by empirical studies[40,67,68]. Consistent with this, the evolution of overproduction has generally been observed in spatially structured environments, which typically support producers. Examples include overproduction of virulence factors by *Pseudomonas aeruginosa*[27,40–42,47], hypersecretion of biofilms by *Vibrio cholerae*[5], *Bacillus subtilis*[39] and *Pseudomonas fluorescens*[15] and even of invertase by *S. cerevisiae*[37,38]. These overproducers can provide key benefits, including enhancing the capacity of the population to repress competitive species by overproducing antibiotics[32], increasing disease virulence and transmission[41,42], or supporting population growth[15,38]. However, even though overproducers evolved in spatially structured environments, they were generally disadvantaged within populations in the long run due to increased costs and exploitation[15,32,39,40].

In contrast, our study demonstrated that *S. cerevisiae* invertase overproduction can evolve and be maintained over 100 generations, even in circumstances where competitors can readily exploit it. This is at odds with numerous evolutionary studies where producers downregulated public goods production in response to competition[17–20,27,31,34–36]. Such prudent regulation can enable producer maintenance[17,18]. Moreover, unlike a study where alterations in public goods regulation differed in the presence and absence of non-producers[31], our overproducers evolved in both scenarios. This suggests that the adaptation was induced by the nutritional environment rather than, for example, detecting the presence of community members through molecular cues[69] or quorum sensing molecules[17,21]. Nevertheless, the overproducers that evolved in the absence of non-producers still gained an advantage over their competitors that produced comparatively less invertase. These lower-level producers can, in this context, be considered exploitative "cheats"[56].

How can invertase upregulation increase producers' competitiveness? By manipulating the nutritional environment and using a newly generated collection of synthetic *S. cerevisiae* strains with modified secondary features of sucrose metabolism (Table 1), we identified two mechanisms driving this unexpected result. Given that invertase (SUC2) is produced in both intracellular and secreted forms[48], we first ruled out that increased competitiveness of overproducers resulted from the concomitant increase in internal sucrose metabolism (Figs. 4b1, c, 5: strain Δagt1 vs. +SUC2Δagt1). Subsequently, we focused on the external sucrose metabolism.

We showed that SUC2 overproduction causes repression of competitors, a key driver of cooperation maintenance[57,58]. This was achieved by reducing the competitor's metabolic efficiency via increasing resource concentrations (Fig. 6), driven by the convex geometry of the rate-efficiency trade-off (Fig. 4e), which is widespread throughout the microbial world[59–61]. Accordingly, overproducers lost their competitive advantage when the influence of the rate-efficiency trade-off was inhibited in low-resource environments[62] (Fig. 5 – 0.05% vs. 1%). We also found that HXT2 was upregulated more by +SUC2 than the wt in 0.05% sucrose (Supplementary Fig. 10e–h), thus ruling out that the extra hexose transporter upregulation by overproducers is only triggered in higher sucrose concentrations. This supports our conclusion that +SUC2 loses its competitive advantage in 0.05% sucrose because the rate-efficiency trade-off is weakened, rather than HXT2 upregulation only being induced in higher sucrose concentrations. Nevertheless, we did not detect a competitive deficit in the overproducers compared to the wt in the absence of the rate-efficiency trade-off, as might be expected considering that overproducers incur additional costs of constitutive overproduction[17]. This led us to conclude that repression of non-producers is not the only mechanism by which overproducers gain a competitive advantage.

Subsequently, we demonstrated that SUC2 overproduction increased the proportion of hexose captured by overproducers compared to the wt (Fig. 7a and Supplementary Fig. 9) by upregulating hexose transporters (Fig. 7b), thereby increasing hexose uptake across the plasma membrane, which is proposed to be the rate-limiting step

of sugar metabolism[49,55]. It was theoretically predicted that producers preferentially secure a fixed proportion of sucrose hydrolysis products that is independent of invertase activity levels and is limited by the properties of the transporters[9]. Our study shows that this proportion is not independent of invertase activity because the increases in hexose from invertase activity induce transporter upregulation, which consequently increases product capture. There is a growing awareness that coordination between the expression of secreted products and their associated uptake mechanisms has important implications for maintaining public good production[70] because it can enable producers to obtain a disproportionate amount of the benefits (i.e., public goods become partially privatised)[6,9,71]. In addition, previous evolutionary experiments with *P. aeruginosa* discovered a sub-population of pyoverdine overproducers in environments that favour non-producers[72]. While the precise causal mechanism was not identified and could relate to pyoverdine's diverse functions[1], findings suggested that non-producers obtained less of the public good benefits (i.e., the public good became more privatised)[72].

We note that it is challenging to experimentally determine the magnitude of the contribution of the two mechanisms - competitor suppression and increased hexose capture - towards the observed overproducer fitness gains because metabolic efficiency is regulated by hexose uptake[62,66], so cannot be easily disentangled.

We also note that the evolved producers in our study were not subjected to genome sequencing. This is because the genetic basis of public-good upregulation is often unclear (e.g., ref. 40) and could relate to various mechanisms, including gene duplication or alterations in complex regulatory systems[49]. Moreover, identifying the genetic basis of the overproducer phenotype was not the aim of this study, as detailed mutational mechanisms were not necessary to demonstrate the causal effect of overproduction on producer fitness. Instead, the causal effects were demonstrated by generating synthetic strains with engineered metabolic features.

While + SUC2 was designed to overproduce invertase, it did so at a relatively higher level than the evolved strains (Supplementary Fig. 4c). However, we do not expect these differences to qualitatively impact our overall findings for the following reasons. Mechanism 1 remains relevant for bestowing fitness advantages to the evolved strains over the wt (Fig. 2a) because invertase-mediated hexose generation, and thus hexose uptake, of the evolved strains is higher than the wt, but lower than + SUC2 (Fig. 4e). Therefore since the rate-efficiency trade-off constrains the efficiency of wt, and this efficiency constraint is intensified for + SUC2 (Supplementary Fig. 2C), similar intensification will hold for the evolved strains with intermediate invertase overexpression, albeit to lesser extent. Mechanism 2 is also relevant since HXT2 expression increases as hexose concentration increases within the relatively low concentrations generated from sucrose hydrolysis (Supplementary Fig. 10a–c). Therefore, even small increases in *SUC2* expression are expected to also upregulate *HXT2*, leading to increased hexose capture by the evolved strains compared to the wt. We thus expect that the fitness advantage of the evolved overproducers when in competition with non-producers would be higher than the fitness advantage of wt but lower than for + SUC2. However, + SUC2 and the evolved overproducers were found to have equivalent fitness advantages over the wt (Fig. 2a c.f. Fig. 5 + SUC2, 1%). This may be because the benefits of extra invertase production by + SUC2 are offset by the metabolic costs of higher, constitutive SUC2 expression[17,18]- which is usually glucose repressed[9]- or the evolved overproducers acquired additional unidentified adaptations, such as in transporters[38,54].

Our study demonstrates that overproduction of public goods can provide direct benefits to producers against exploitative competitors, in this case by suppressing competition and enhancing product capture. The latter is accomplished via the interaction between two traits, namely catabolic enzyme secretion and sugar transporter expression, which enables public good production to persist, as well as amplify

population growth rates. This finding exemplifies how pleiotropy can stabilise cooperation[23,73] and highlights how multiple trait interactions can influence the evolution and functions of microbial communities in non-trivial ways[74].

Separating the competitive versus cooperative benefits of traits is not straightforward, and our interpretation is that public metabolism via secreted products provides both cooperative and competitive benefits. Although secreted products can be used by other cells, it has been argued that their production might have evolved as a result of direct benefits to the producer, making them a competitive trait, rather than cooperative[22,43,56,75]. The competitive nature arises through partial privatisation, which, as discussed above, can arise through spatial structure, but it can alternatively result from specific uptake receptors[43], or in this case, from SUC2 being localised at the cell surface[49]. The identification of invertase non-producers and *SUC* copy number variations amongst "natural" strains supported the premise that invertase secretion is cooperative[8,76]. However, an alternative explanation suggests that strains adapt to differing sucrose environments, rather than social interactions. This is partly due to the lack of invertase non-producers amongst natural strains of *S. paradoxus*[75]. Our study demonstrates how producers can tune metabolic processes, including substrate degradation and uptake capacities, to diminish non-producers' potential benefits, which might explain why SUC2 non-producers are seemingly rare in nature[75,76].

Beyond the benefits that SUC2 overproduction can provide to overproducers from competitor suppression and enhanced product capture, what other factors influence the evolution of overproduction? While the overproduction of SUC2 has been experimentally evolved previously[37,38], it occurred in spatially structured environments and so was associated with adaptation to the nutritional environment rather than enhancing fitness against exploitative non-producers. Even when non-producers are at a selective advantage locally, overproducers could also evolve when they form more productive and resilient populations (Fig. 7c)[77]. This can also help survival through population bottlenecks as they disperse[46], or increase transmission to new hosts[41]. However, increased growth rates of overproducers can coincide with reduced efficiency (Supplementary Fig. 2c). Therefore, although spatial structure could favour overproducers by limiting exploitation from non-producers, it also selects for efficient metabolism[44,59], meaning that spatial structure could hypothetically both promote and hinder the evolution of overproduction due to the rate-efficiency trade-off.

The use of public goods to acquire nutrients is widespread[1-4], despite the threat of exploitation[5,6,8-14], which could stem from it being the quickest way of obtaining resources[46,78]. Yet, in this study, we demonstrate how producers can actually enhance their fitness by increasing the production of metabolic secretions. Overproduction can increase both competitiveness against neighbours, and the growth rate of the overall population. This finding is important because the microbial cooperative breakdown of polymeric carbohydrates is key for carbon cycling in all ecosystems, with roles in the human gut[28,79], soil[80], oceans[5,81] and during infection[2,4]. Understanding the evolution of public goods, the production of hydrolytic enzymes, and the subsequent consequences for microbial population functioning is, therefore, important for sustainably managing health, disease, climate change and wildlife conservation.

## Methods
### Strains
Details of all strains used in this study are shown in Supplementary Data 3. Strains had identical auxotrophic requirements to strains they were being compared with or competed against. The wild-type "producer" strain was based on the wild-type *MAL*-constitutive strain CEN.PK2-1C[82] (*MATa, leu2-3_112, ura3-52, trp1-289, his3-Δ, MAL2-8ᶜ, SUC2, hxt17Δ*). The "non-producer" was CEN.PK2-1C that had its *SUC2* gene replaced with the gentamycin resistance marker, *kanMX*

(*suc2Δ::kanMX*) (Supplementary Fig. 1). These strains expressed the selectively neutral[46] fluorescent proteins, *mCherry* (producers) or *eYFP* (non-producers), from the strong, constitutive *TEF1* promoter so they could be readily identified during co-inoculation (Supplementary Fig. 14). This construct was introduced into the *ura3-52* allele of the *URA3* locus using the integrating plasmid pRS306 that was linearised with *EcoRV*. Tryptophan prototrophy was restored in the non-producer and wt producer by integrating the empty pRS304 plasmid that had been linearized with *EcoRV* into the *trp1-289* allele of the *TRP1* locus.

The overproducing strains (+ SUC2 and + HXT7) were generated in the wt producer genetic background. However, instead of restoring tryptophan prototrophy with the empty pRS304 plasmid, over-expression and tryptophan prototrophy were achieved by introducing a second copy of the respective genes that were driven by the strong, constitutive *GPD* promoter using the integrating plasmid pRS304, linearised with *EcoRV* for *SUC2* and *XbaI* for *HXT7*, into the *trp1-289* allele of the *TRP1* locus. Δagt1 and + SUC2Δagt1 were generated by the replacement of *AGT1* with *kanMX*. Gene replacement was confirmed by PCR (Supplementary Fig. 6). The *AGT1/SUC2* double mutant was generated in the CEN.PK2-1C background by replacing the *AGT1* gene with *TRP1* to restore tryptophan prototrophy (*agt1Δ::TRP1*), and replacing *SUC2* with *kanMX* (*suc2Δ::kanMX*), both of which were confirmed by PCR (Supplementary Fig. 7d).

TM6* is a mutant of CEN.PK2-1C has only a single hexose transporter gene[66]. TM6* was made uracil auxotrophic by introducing the *ura3-52* allele into the *URA3* locus and counter-selecting with 5-fluoroorotic acid (Melford Laboratories – F10501). TM6*Δsuc2 had its *SUC2* gene replaced with the gentamycin resistance marker (*suc2Δ::kanMX*)[44]. TM6* and TM6*Δsuc2 constitutively expressed *mCherry* or *GFP*, respectively, from the *TEF1* promoter, integrated in the *ura3-52* locus using the pRS306 plasmid, linearised with *EcoRV*, restoring uracil prototrophy. TM6* + SUC2 was generated by firstly making TM6* tryptophan auxotrophic by introducing the *trp1-289* allele into the *TRP1* locus and counter-selecting with 5-fluoroanthranilic acid. Into the resulting strain, the same construct used to generate + SUC2 was introduced.

*HXT2* expression was measured by introducing a construct that contained *HXT2*-promoter-driven *eGFP* (*pHXT2-eGFP*) into the *URA3* locus of non-fluorescing strains of the wt producer and + SUC2. Approximately 800 bp upstream of the *HXT2* start codon was cloned with *eGFP* and the *PMA1* terminator into the integrating plasmid pRS306 that was linearized in the *URA3* locus with *EcoRV*.

### DNA analysis and genetic modification
Nucleotide sequences were retrieved from the *Saccharomyces* Genome Database[83] (version R64-2-1). PCR was performed with CloneAmp HiFi PCR premix (Takara Bio - Cat. No. 639298) for genetic modifications and SapphireAmp Fast PCR Master Mix (Takara Bio - Cat. No. RR350A) for colony PCR. Plasmids were generated with an In-Fusion HD Cloning kit (Takara Bio – Cat. No. 638918). Primer nucleotide sequences used in this study are shown in Supplementary Data 4. Southern blot analysis (Supplementary Fig. 1) was conducted using digoxigenin(DIG)-labelled probes (Roche Applied Science - Cat. No. 11585550910), visualised with CDP-*Star* Chemiluminescent Substrate (Sigma Aldrich - Cat. No. C0712).

### Growth, evolutionary and competition experiments
Experiments were conducted in synthetic complete (SC) media (6.9 g/l yeast nitrogen base without amino acids, 790 mg/l complete supplement mixture (Formedium – Cat. No. CYN0410, DCS0019)) supplemented with the indicated concentration of sucrose or D-glucose (Fisher Scientific). Media was filter sterilised (0.2 μm) to remove particulate matter that interferes with flow cytometry and avoid sugar breakdown. Starter cultures were grown overnight in 4 ml SC media,

1% glucose (except for *pHXT2-eGFP* experiments where 1% D-galactose (Fisher Scientific) or 4% glucose was used), at 30 °C with 180 r.p.m. orbital shaking. Cells were collected by centrifugation, washed and resuspended in an equal volume of fresh media containing the appropriate concentration of sucrose or glucose and diluted to $10^2$ CFU/μl.

Growth, evolutionary and competition experiments were conducted in 48-well suspension culture plates (Greiner Bio-One – Cat No. 977102) inoculated with $10^2$ CFU/μl in a volume of 640 μl. Initial populations were established from common starter cultures of each genotype before being inoculated into individual wells, which were each considered as a replicate. Plates were sealed with a 50 μm thick polyester film (VWR – Cat. No. VWRU60941-120) to minimise evaporation and prevent cross-contamination, which was pierced at the edge above of each well with a sterile needle for gas exchange. Uninoculated blanks were included as contamination checks. Microplates were incubated at 30 °C in a microtitre plate reader (FLUOstar Omega (BMG Labtech) for competition and evolutionary experiments; Spark 10 M (Tecan) for growth assays) with maximum orbital shaking (700 r.p.m. – FLUOstar Omega; 510 r.p.m. for Spark 10 M).

For growth assays, the population density was measured in the plate reader by absorbance (620 nm) every 20 minutes and converted to CFU/μl using previously established calibrations[46]. Since the plate reader does not accurately measure population densities below ~ $5 \times 10^2$ CFU/μl in these conditions, precise initial population densities were measured by flow cytometry. Metabolic efficiency is estimated as population yield (i.e., carrying capacity ($K$) per unit of sugar in the media (%) – taken at the indicated time points when sugar has been exhausted).

For competition and evolutionary experiments, mixed strain populations were initiated by mixing strains at the different initial frequencies, which were established by combining volumes of approximately equal density axenic cultures at the appropriate portions. Initial frequencies of strains were 0.5, except for the following circumstances: (1) frequency-dependence tests where each of the initial frequencies are indicated on the plot axes/legend. (2) During evolutionary experiments where mixed populations were initiated with a minority of producers (approximately 10%) so that they did not have a higher chance than non-producers of acquiring adaptive mutations in non-social traits by being numerically dominant, as found previously[25,26,53]. (3) During multi-season competitions to test resistance to invasion by non-producers, where producer initial frequency was ~ 90% (Fig. 3b). Precise initial cell densities and genotype frequencies were measured by flow cytometry (Guava easyCyte 10HT System using Guava InCyte software version 3.2) with genotypes distinguished by their fluorescent protein properties (Supplementary Fig. 14). Growth proceeded for 24 h, by which time populations had reached an approximate stationary phase (Supplementary Fig. 2a), except for the following treatments where an extended period of time was required due to slower growth caused by low resource concentrations (Supplementary Fig. 8a) or slow-growing genotypes (Supplementary Fig. 11). Extended incubation times were required for: (1) The competitions in 0.05% sucrose where cultures were grown for 28 h. (2) The competition with TM6*-based strains, where cultures were grown for 42 h. At the end of the growth season, changes in genotype frequency were measured by flow cytometry. For evolutionary experiments, at the end of each season (24 h), cells were diluted 600x into fresh SC media (1% sucrose) to return to densities of ~ $10^2$ CFU/μl and re-inoculated into a fresh microplate for the subsequent season.

Growth rates at very low population densities in 1% sucrose media (Supplementary Fig. 9a) were also conducted in the same 48-well plates and incubated/shaken in a microplate reader (FLUOstar Omega), but both initial and final (after 4.7–7 h) population densities were measured by flow cytometry because population

densities were below the detection limit of absorbance readings (620 nm).

## Measurements of growth rate, relative fitness and normalised relative fitness

Growth rates were calculated as Malthusian growth parameters ($w$)[52]:

$$w = \frac{\ln(N(final)/N(initial))}{t} \quad (1)$$

where $N(initial)$ is the starting population density and $N(final)$ is the density at time $t$ (h). For growth rates in Supplementary Figs. 2, 7, 8 and 11, $t$ was taken as the time point at which population density exceeded $8 \times 10^3$ CFU/μl in order to capture at least 6 doubling, variations in lag, and/or density-dependence[46], and to precede substantial growth deceleration as resources become limiting. For growth rates in low glucose concentration, $t = 1.385$ h to precede glucose depletion and diauxic shifts (Supplementary Fig. 9c–e), while at low population densities (Supplementary Fig. 9a), $t$ was taken as 4.7–7 h to precede substantial environmental hexose accumulation from extracellular sucrose hydrolysis.

Relative fitness $v$, for a replicate $i$, of a given genotype was calculated as previously[33]:

$$v_i = \frac{x_2(1-x_1)}{x_1(1-x_2)} \quad (2)$$

where $x_1$ and $x_2$ are the initial and final frequency, respectively, of that genotype.

For ease of interpretation and to normalise results between different experiments, a normalised relative fitness $S_i$, was calculated for each replicate $i$ of evolved or engineered SUC2/HXT7-overproducing strains whereby the relative fitness of each replicate, when in competition against their corresponding non-producers, $v_i$, was normalised against the mean relative fitness of the corresponding wt when in competition against the same non-producers, $\bar{v}$, (see Supplementary Data 3 for strain details). This leads to the following formulation:

$$S_i(focal\ strain) = \frac{v_i(focal\ strain)}{\bar{v}(corresponding\ wt)} \quad (3)$$

where,

$$\bar{v}(strain) = \sum_{i=1}^{n} v_i(strain),$$

and $i = 1..n$.

For competitions with the TM6*-based strains in Fig. 5, the normalised relative fitness $S_i$ is defined slightly differently. Namely, due to variation in competitiveness of the strains in the non-social environment (i.e., in 1% glucose – Supplementary Fig. 15), in Eq. (3) the relative fitness of TM6* + SUC2 strain, $v(TM6^* + SUC2)$, was normalised against its mean relative fitness in 1% glucose, $\bar{v}^g(TM6^* + SUC2)$, while the mean relative fitness of the corresponding wt, $v(TM6^*)$, was normalised against its mean relative fitness in 1% glucose, $\bar{v}^g(TM6^*)$, to get

$$S_i(TM6^* + SUC2) = \frac{v_i(TM6^* + SUC2)/\bar{v}^g(TM6^* + SUC2)}{(\bar{v}TM6^*)/\bar{v}^g(TM6^*)} \quad (4)$$

Note that competition experiments with TM6*-based strains were also conducted 4 times (rather than 3 times for the CEN.PK2-1C-based strains) with 1% glucose competitions run in parallel to the 1% sucrose competitions.

The Eqs. (3) and (4) can be interpreted in the following way: if $\ln(S_i) > 0$, the focal strain is at a selective advantage, while if $\ln(S_i) < 0$,

the focal strain is at a selective disadvantage compared to its corresponding wt. If $\ln(S_i) = 0$, the fitness of the focal strain when in competition with a non-producer is equal to the fitness of its corresponding wt when in competition with a non-producer (see Supplementary Data 3 for the list of strains).

## Enzymatic assay of secreted invertase activity

Secreted invertase of live *S. cerevisiae* cells was measured using a colourimetric assay that detects generated reducing sugars using 4-Hydroxybenzhydrazide (Sigma-Aldrich). Cells were grown overnight in 1% glucose SC media, washed, and transferred into 0.01% glucose SC media and incubated at 30 °C, 180 r.p.m. for 2.5 h to de-repress *SUC2* expression[9]. Cells were washed and resuspended in 0.1 M sodium acetate (Fisher Scientific), pH 4.5. Cells ($2.13 \times 10^6$ in 100 µl) were combined with 900 µl 1% sucrose in 0.1 M sodium acetate, pH 4.5 (or 4% sucrose where specified) and incubated at 30 °C for 15 min. Invertase activity was detected in this mixture as the concentration of reducing sugars in the supernatant that were released from sucrose hydrolysis: 100 µl was transferred to 2.9 ml 0.5% (w/v) 4-Hydroxybenzahydrazide in 0.5 M NaOH and heated at 100 °C for 5 min, then cooled to room temperature and absorbance was measured at 410 nm. Measurements were made against un-inoculated blanks and converted to hexose concentration using a calibration of known glucose concentrations (treated as equivalent to fructose). Assays were conducted in triplicate, with experiments repeated twice. For the six evolved strains, in each experiment, the invertase activity of two evolved strains was measured concurrently with the ancestor (full replicate sets shown in Supplementary Fig. 3b) so that the invertase activity of the evolved strains could be normalised against it. Environmental hexose concentrations from the supernatant of batch cultures (Fig. 6a) were equivalently measured using 4-Hydroxybenzhydrazide.

## Estimating *HXT2* expression

Cells were pre-cultured overnight in 4 ml of either 1% galactose SC media or 4% glucose SC media. Cells were washed and resuspended in SC media lacking a carbon source before being inoculated into each induction condition at a density of approximately $10^2$ CFU/µl. 640 µl of each culture was incubated at 30 °C in 48-well suspension culture plates (Greiner Bio-One) with 700 r.p.m. orbital shaking in a FLUOstar Omega plate reader for 3 h. *eGFP* expression was measured by flow cytometry using a blue excitation laser ($488 \pm 5$ nm) and detecting emission with green (525/30 nm) (Guava easyCyte 10HT System using Guava InCyte software version 3.2) with 4388-5000 events measured for each genotype/condition combination. *eGFP* measurements, reported as relative fluorescence units (RFU), were normalised against the median value of the non-fluorescing CEN.PK2-1C strain that was incubated in identical conditions.

## Data analysis

Statistical analysis was conducted with R version 4.0.4. Pairwise comparisons were made with a two-sample, two-sided Student's *t* test. If data had significantly different variances, as measured by the F-test for homogeneity of variance, then a Welch's *t* test was used. Larger datasets were analysed using general linear models (GLM) (or generalised linear model for Fig. 5) with response variables (vertical axes) tested against the plotted explanatory variable (horizontal axis) with two-way interaction terms between multiple explanatory variables. These terms include interactions between the following explanatory variables: genotype:sugar type (Fig. 2), genotype:frequency (Figs. 3a, 6c), genotype:season (Fig. 3b) and genotype:sugar concentration (Fig. 5). Multiple pairwise comparisons were conducted with post-hoc Tukey multiple comparisons of means (Honest Significant Difference (HSD) method) with an analysis of variance model.

## Reporting summary

Further information on research design is available in the Nature Portfolio Reporting Summary linked to this article.

## Data availability

The data supporting this article can be found at https://doi.org/10.6084/m9.figshare.26206334. Source data are provided as a Source Data file, and uncropped gel and blot scans can be found in the Supplementary Information. Source data are provided in this paper.

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

## Acknowledgements
We thank Robert Beardmore, Lisa Butt and Emily Dickens for their comments and useful discussions. We thank Stefan Hohmann and Peter Dahl (University of Gothenburg) for providing yeast strains. R.J.L. is funded by a Biotechnology and Biological Sciences Research Council-National Science Foundation/BIO grant (BB/T015985/1) to I.G., P.J.H. is funded by a Leverhulme Research Project Grant (RPG-2019-238) to I.G.

## Author contributions
I.G. and R.J.L. conceived the idea, designed the experiments, analysed the data and wrote the manuscript; R.J.L., P.J.H. and M.H. conducted the experiments.

## Competing interests
The authors declare no competing interests.
