## [Peer Review File · Nature Communications]

Experimental evolution of yeast shows that public-goods upregulation can evolve despite challenges from exploitative non-producersReviewers' Comments:

Reviewer #1:

Remarks to the Author:

In this study, the authors tested whether overproduction of a public good can lead to benefits to the producer. Contrary to well-founded expectations, they convincingly showed that this can occur in *S. cerevisiae*. In particular, they argue that private benefits to producers become larger as the producer generates more of the public good. Overall, I found the writing and logic to be clear and easy to follow (with one notable exception highlighted in the next paragraph). I felt the authors conducted a series of elegant experiments that thoroughly support their conclusions. This study would be a valuable contribution to the literature.

I had one major concern that needs to be addressed before publication. I had a lot of trouble following the authors description of the efficiency trade-off and how this relates to their results. Relatedly, I thought the phrasing about suppressing competitors was often vague and needed more explanation. On lines 200-209, the authors first provide details about how competition can be suppressed. I struggled quite a bit with this framing. In particular, lines 206-208 make it sound as though this is simply a private benefit (i.e., you get to keep some of what you produce). That argument would make sense to me. However, I'm struggling to relate this simple private benefit argument with what seems to be a more complicated argument about metabolic tradeoffs. Are such tradeoffs really necessary or is it simply a matter of getting the "first shot" to uptake what is produced (i.e., private benefits). I list this as a general comment that affected my ability to follow some of the big-picture arguments and conclusions of the study, but specific instances where I found this problematic are listed below under specific comments.

Specific comments:

Line 52: This is the third time that competitor suppression is presented as an option, but I'm still not clear on what it means. Looking at reference 43 (admittedly quickly), I'm struggling to see how this is distinct from what is mentioned as the other benefit: "increasing access to resources." Please make sure terms are defined.

Lines 110-120: Very clear and elegant argument.

Lines 131-132: I'm struggling to follow the logic here. Isn't it possible that the evolved producers enhanced sucrose, but not glucose, uptake?

Lines 170-171: Again, I'm unclear on how it would suppress exploitation?

Line 217: The authors use carrying capacity to measure metabolic efficiency. More justification and explanation of this choice is needed.

Lines 220-222: I see why this reduced efficiency would exist, but shouldn't efficiency also be

reduced for the producer with higher hexose concentrations?

Lines 234-241: In this chunk of text, fitness should be taken to mean “fitness relative to non-producers”, right? That is, saying +SUC2 is expected to have lower fitness than the wild type is meant to say + SUC2 is expected to have lower fitness compared to non-producers than the fitness of the wild type compared to non-producers. This point should be made explicitly, I struggled to follow the logic here at first, because I was not thinking of the appropriate measure/comparison for fitness.

Lines 365-366: Thinking it over again, is it possible that the uptake of sucrose is upregulated as a threshold effect (i.e., more transporters are only produced given a sufficiently high sucrose concentration)? If this is the case, then perhaps +SUC2 loses its competitive advantage at low sucrose concentrations not because of an efficiency tradeoff, but rather because sucrose concentrations are not high enough to trigger the increase in uptake efficiency. Is there an argument that could rule out this alternative explanation?

Lines 389-395: The results from Scott (2022, <https://doi.org/10.1073/pnas.2214827119>) appear relevant to this paragraph.

Lines 420-423: A very nice point!

Discussion: There have been a number of theoretical studies that consider how partial privatization can lead to the evolution of public goods production, but the authors only cite Gore et al (2009). It might be nice to try to make a few more connections between the results of this study and previously developed theory. See, for example, Inglis et al (2016, <https://doi.org/10.1098/rspb.2015.2682>), Estrela et al (2016, <https://doi.org/10.1111/1462-2920.13028>), and Lerch et al (2022, <https://doi.org/10.1371/journal.pcbi.1010666>).

Line 553: Is there a reference to this definition of a selection coefficient? To me, this looks more like a relative fitness (fitness of one type divided by fitness of the other). In contrast, selection coefficients (at least in the realm of population genetics) tend to have the form (relative fitness of one type) / (relative fitness of the other type) – 1 (see, for example, Otto and Day Chapter 3). I understand that it provides the relevant information, so this is only a semantic concern, but it is good to keep consistent terminology with past literature when possible.

Finally, I’d like to make some suggestions that the authors may find useful but should not preclude publication. In many cases, these are recommendations for how to make the writing clearer (in my opinion).

Abstract: I found that the description of the study and results was somewhat brief given how much background was provided. For example, no information is provided about invertase. Further, neither “competitor suppression” nor “increasing product capture” are completely self-explanatory but these are key to the results. In contrast, lines 12-22 provide quite a bit of detail before mentioning

the present study. The sentence on lines 14-16 is of course nice motivation but I wonder if it's needed in an abstract that feels rushed on the details of the study.

Lines 77-78: The authors provide a useful summary of their study at the end of the introduction. This sentence is rather vague, however. It would be nice to tell the reader in a bit more detail why overproduction can evolve in this experiment even in competition with non-producers. As written, this chunk feels like a bit of a cliffhanger.

Lines 93-108: As someone that is not knowledgeable about specific mechanisms for microbial metabolism, this paragraph was challenging to understand. It would be clearer for readers that are uninformed about MAL, SUC2, etc., if the authors "flipped" this paragraph. That is, begin by explaining to the reader that in past studies the non-producer is dependent on the producer, which will not be the case in this study. With that background, the mechanistic/genetic details would be easier to follow.

Reviewer #2:

Remarks to the Author:

This study investigates the evolution of public goods production under conditions typically leading to production decline due to selection pressure by non-producing 'cheats'. Using invertase production in *Saccharomyces cerevisiae* as a model trait/system, the authors find that – contrary to previous research – invertase over-producers evolve even in the presence of non-producers. Using a combination of different growth conditions and synthetic *S. cerevisiae* strains with modified secondary features of sucrose metabolism, the authors identify two mechanisms, competitor suppression and increased product capture, which keep non-producers at bay and allow the evolution of overproducers.

In general, the work is well written, and the methods and results sections are robust and supported by the data presented in the figures. However, I have a general comment and a number of (mostly minor) concerns that prevent me from recommending publication in its current form. My general concern relates to the approach of testing synthetic strains rather than evolved clones to identify mechanisms favoring the evolution of over-producers. In LL 383-388 of the discussion, the authors explain their reason for not following up their results with genome sequencing ('genetic basis of public-goods upregulation often unclear'). I partly sympathize with this view – but I still would have liked to see some screening of obvious candidates (such as the [promoters of] genes modified in the synthetic strains). At the very least, I think that the authors should discuss potential differences between their engineered over-producer vs. the evolved clones (for instance regarding the extent of invertase up-regulation [not yet statistically compared?]) – and how these differences would change the importance of the two mechanisms favoring over-production. Indeed, it might be worthwhile to consider introducing the general approach (i.e., testing synthetic strains instead of evolved clones) early on. I found it to be a strength of this study (and readers wouldn't go through the greater part of the manuscript expecting sequencing data).

Please find below a number of detailed comments that will hopefully help to further improve the manuscript.

Detailed comments:

Title: Consider removing the first part of the title ('Beating the cheat with kindness'). I think the second part is clear and straight to the point, and I find the mentioning of "kindness" somewhat misleading. After all, upregulation of public-goods production seems to evolve due to direct ("selfish") fitness benefits to producers, not due to indirect benefits (e.g., through altruism).

LL 30-31: I suggest replacing the second 'products' by sth like 'metabolite' to make it clearer which products you are referring to in the following sentence.

L 39: Consider briefly explaining what 'prudently regulating expression' would look like.

LL 46-56 (and LL 86-92, LL 345-347): The distinction between what you consider strong versus weak selection pressures from non-producers is not entirely clear to me. Do you consider strong selection pressures only to occur if non-producers can outcompete producers? In that case, I would suggest abandoning the strong/weak selection differentiation altogether, focusing instead on the different competitive outcomes (non-producer winning = strong selection [and narrow-sense cheating]; non-producer doing better, but still losing = weak selection). Irrespective of this point, I also suggest that you make a clear distinction between (up)regulation (plastic response) and the evolution of upregulation (generally increased production). The references cited in L 48 contain examples of both (ref 17 vs 18).

LL 54-55 (& 78-80): This is generally true, but this study (<https://doi.org/10.1111/mec.16119>) reports that (a few) hyper-producers of pyoverdine evolved under conditions favoring cheating on this siderophore in *Pseudomonas aeruginosa*.

L 73: Based on the pattern in Fig 1B, I would argue that they prevented fixation of non-producers, but not their invasion (frequency of producers declined over evolutionary times).

L 78: 'prevent' instead of 'prevents'; missing 'from' after 'them'?

L 91: Consider adding a 'commonly used' or 'previously used' before 'S288c genetic background' for clarity.

L 93 (& L 97): 'locus' instead of 'loci'?

LL 93-108: I would find this paragraph easier to follow if you began by stating that the previously used S288c does not have an alternative way of metabolizing sucrose, and that you therefore constructed a suc2-deletion mutant in another background that had additional means to metabolize sucrose (MAL12 + AGT1), making it a stronger competitor (as is, the point of the paragraph only becomes apparent half-way through). Also, I suggest specifically stating here that you used a wild-type with the same background (not S288c) as producer.

LL 105-108: What is a season, and how long does it last? I'm also wondering about your differentiation between 'coexisting' (S288c producer and non-producer) and 'outcompeting' (CEN.PK2-1C producer and non-producer): Fig 1A shows that the CEN.PK2-1C non-producer outcompetes the producer, but it does not show that the producer went extinct during one season. Unless this was the case, I would argue that they still co-existed over the duration/at the end of that season. Please clarify.

Figure S2: Please describe the strains you used, e.g. in the figure legend (the reader will not have encountered some of the strains [+HXT7, +SUC2] when reading the manuscript and the supplements in parallel).

Figure 1: Please indicate in the legend which initial wt frequencies were tested (Fig. 1A) – it is impossible to read out the exact frequencies for the four lowest levels from the graph. Please also indicate the number of replicates per frequency (they seem to vary between 3 and 4?). In Fig. 1A,

consider showing the regression line as a dashed/dotted line, which would make clearer that the relationship was not significant. Finally, the part of the legend referring to Fig.1B could be clearer regarding what's actually shown in the graph (e.g., 'b The frequency of producers initially declined, but then plateaued during a long-term evolution experiment). Finally, I would not consider an evolution experiment long-term, if it only runs for 10 cycles (also relevant for L 161).

LL 110-112: Please indicate the number of independent lines/replicates of your evolution experiment.

LL 112-114: Please indicate initial relative frequency of the producer (from Fig 1B, I gather that it was 10%) and describe the decline (e.g., 'from x to y % of the population') so that readers don't have to jump to the figure/methods. Along the same lines, please also explain here why you used this (relatively low) initial frequency (and why you used a much higher frequency in the experiments described in LL 160-162 and depicted in Fig 3B?).

LL 121-123: Is that one clone per line? And how many control lines did you run? Also, three?

Figure 2: Looking at Fig. 2A, I'm wondering how the evolved producers did when compared to the ancestral non-producer instead of the wt? Did they have a higher fitness, and was this dependent on whether they had evolved with a non-producer vs. alone?

L 144: Please briefly indicate here how invertase activity was measured (I'm not suggesting to provide any details, mentioning the nature of the assay is enough).

LL 153-154: Fig.3A seems to suggest that +SUC2 actually wins in direct competition against the non-producer. Consider indicating that here.

Figure S4: Typo in second line (activity)

Figure 3: Please describe the strains in the figure legend. The reader will not yet have encountered +HXT7 when Figure 3 is first referenced.

Figure 4: You describe how i) and iii) were tested, but what about ii)? Consider describing that here, too. Please also consider changing your numbering of the different parts of the graphs; using i), ii), etc. both for numbering the hypothesis and referencing specific parts of the figure is confusing.

LL 187-188: This sentence is hard to understand, consider rephrasing.

LL 193-197: To me, the result that 'agt1-deletion strains did not prevent the invertase overproducer (+SUC2 Δ agt1) having a selective advantage over its equivalent wt producer (Δ agt1)' shows that the enhanced competitiveness of the SUC2 overproducers is not ONLY caused by increasing internal sucrose metabolism. But I don't understand how you can exclude the possibility that it partly explains the increased competitiveness. To me, this would require directly comparing the magnitude of the respective competitive advantages. Please clarify.

LL 264-265: Consider briefly giving the gist of the approach.

L 281: I suggest replacing 'this increase' with 'this qualitative difference' (in HTX2 expression of +SUC2 vs. wt) for clarity.

L 336: Consider adding an 'only' after 'overproduction' for clarity.

LL 343-344: Also when populations grow/evolve in spatially-structured environments?

L 373: Replace 'producers' with 'over-producers' (the wt is also a producer).

L 389-390: Do you mean it is the first that shows benefits of overproduction in a situation where non-producers can outcompete the wt?

L 404: 'it' = 'invertase production'?

L 410: I suggest adding 'invertase' before 'non-producers' – non-producers of other 'public goods' such as siderophores are not necessarily rare (e.g., <https://doi.org/10.1038/s41467-017-00509-4>)

L 492-493: Did you inoculate different wells/replicates of the same treatment with different precultures (such that, say, different replicates of +SUC2 monoculture growth were started from different precultures)?

Reviewer #3:

Remarks to the Author:

This paper is a strong investigation into the effects of invertase overexpression in *S. cerevisiae*. It is a tour-de-force that dissects exactly how overexpression of SUC2 can increase fitness, and it beautifully shows that it is through increasing the amount of glucose available in the external environment (which has the effect of decreasing metabolic efficiency of competitor cells) and through increasing the amount of hexose transporters for private glucose utilization. I have very little to contribute to the experimental work, save a minor comment (see below).

The main issue I have with this paper is the framing of the results. It seems that overexpression of invertase is the result of selection due to sucrose in the environment and not the result of the presence of non-producers. Figure 2 clearly shows that strains increase invertase expression when evolved in the presence of sucrose without non-producers. Thus, while the rest of the experimental results show the effect that increased invertase expression can have on non-producers (i.e., decreased metabolic efficiency at certain sucrose concentrations), that is not why it is selected/favored in the first place. These effects are by-products of selection for utilizing sucrose. Lines 26-28 in the Abstract are not quite accurate: "A combination of competitor suppression and concurrently increasing product capture were revealed to be driving the evolution of overproducers, in environments originally thought not to support public goods production." Competitor suppression is not driving the evolution of overproducers because it occurs without non-producing competitors. The most likely case is the direct, private benefits that the cells receive that are driving the increase in invertase production. Since this particular trait has such a strong private component, it makes the results less about the evolution of public goods expression. I think the authors should revisit the framing of the paper.

Minor comments:

Figure 3b: Why is there such a big difference in the initial proportion of producers in the transfer experiments in 1b and 3b? Also, in Figure 1b, the proportion of wt producers seems to level-off, which the authors suggest is solely due to adaptation on the part of the producers. Why don't the wt producers also reach a plateau in Figure 3b (it is the same amount of transfers). Is it possible that the results in Figure 1b would be better interpreted as a combination of an increase in fitness, as well as frequency dependent selection?

Typos: Line 93 Not sure if the authors would like this to be singular or plural. I think plural, so remove "an".

Line 152: When you say “on” a certain medium, it usually indicates agar-based solid medium. Better to replace with “in” for liquid.

Reviewer #4:

Remarks to the Author:

The authors of the study used the yeast *Saccharomyces cerevisiae* as a model system to challenge the conventional understanding of microbial production of public goods. Contrary to past studies, they found that overproducers can evolve even under strong selection pressure from non-producers. This study claims to have identified novel mechanisms, including competitor suppression and increased product capture, as driving the evolution of overproducers.

I have read other papers from the group and found the results appealing. However, unlike the other papers, which were easy and pleasant to read, this manuscript was very difficult to follow, from the abstract to the main text.

- The general writing lacks a logical flow that is capable of guiding the reader through the main findings.
- Abstract needs to be restructured; it has a very long introduction that doesn't seem to be necessary while also doesn't communicate the relevance of the work.
- The introduction seems to be repetitive (first two paragraphs) and doesn't present a clear narrative.
- The figures and captions should provide a clear take-away.

Unfortunately, I do not recommend the publication of this manuscript in its current form. However, I encourage the authors to restructure the findings (text and figures) to tell a better story that may be interesting to a broader audience.

We would like to thank the reviewers for their time and thoughtful comments. We have comprehensively addressed all the queries raised, including adding new experimental data (S.Fig 5b, S.Fig 10e-h), analysis (S.Fig 3a, S.Fig 4c, Fig. 5/S.Table 2) and schematics (Fig. 4e). Please see below our point-by-point responses.

** If you wish to forward this email to your co-authors, please delete the link to your author home page below **

Dear Professor Gudelj,

Thank you again for submitting your manuscript "Beating the cheat with kindness: public-goods upregulation can evolve despite challenges from exploitative non-producers" to Nature Communications. We have now received reports from 4 reviewers and, after careful consideration, we have decided to invite a major revision of the manuscript.

As you will see from the reports copied below, the reviewers raise important concerns and we ask you to address them with additional work. Without substantial revisions, we will be unlikely to send the paper back to review. In particular, please keep in mind that we can be flexible with formatting limitations to allow you to fully address referee concerns.

If you feel that you are able to comprehensively address the reviewers' concerns, please provide a point-by-point response to these comments along with your revision. Please show all changes in the manuscript text file with track changes or colour highlighting. If you are unable to address specific reviewer requests or find any points invalid, please explain why in the point-by-point response.

REVIEWER COMMENTS

Reviewer #1 (Remarks to the Author):

In this study, the authors tested whether overproduction of a public good can lead to benefits to the producer. Contrary to well-founded expectations, they convincingly showed that this can occur in *S. cerevisiae*. In particular, they argue that private benefits to producers become larger as the producer generates more of the public good. Overall, I found the writing and logic to be clear and easy to follow (with one notable exception highlighted in the next paragraph). I felt the authors conducted a series of elegant experiments that thoroughly support their conclusions. This study would be a valuable contribution to the literature.

Thank you

I had one major concern that needs to be addressed before publication. I had a lot of trouble following the authors description of the efficiency trade-off and how this relates

to their results. Relatedly, I thought the phrasing about suppressing competitors was often vague and needed more explanation. On lines 200-209, the authors first provide details about how competition can be suppressed. I struggled quite a bit with this framing. In particular, lines 206-208 make it sound as though this is simply a private benefit (i.e., you get to keep some of what you produce). That argument would make sense to me. However, I'm struggling to relate this simple private benefit argument with what seems to be a more complicated argument about metabolic tradeoffs. Are such tradeoffs really necessary or is it simply a matter of getting the "first shot" to uptake what is produced (i.e., private benefits). I list this as a general comment that affected my ability to follow some of the big-picture arguments and conclusions of the study, but specific instances where I found this problematic are listed below under specific comments.

We apologise for not making the section on rate-efficiency trade-off and how it relates to our results clearer. We have now addressed this by providing more detail when describing Mechanism 1 (lines 82-86 and 394-396) and by including a more detailed schematic in Fig. 4e (an earlier version of this was Supplementary Fig. 8a in the original submission) that illustrates the role of a convex trade-off curve in enabling producers to suppress their competitors. We also provide detailed responses to the general comments below relating to this point.

Specific comments:

Line 52: This is the third time that competitor suppression is presented as an option, but I'm still not clear on what it means. Looking at reference 43 (admittedly quickly), I'm struggling to see how this is distinct from what is mentioned as the other benefit: "increasing access to resources." Please make sure terms are defined.

We have now clarified that upregulation of public goods in spatially structured environments has previously been shown to "suppress interspecific competitors with antibiotics (41) or non-exploitable siderophores (43)." (LL 54-55). We have also expanded the earlier descriptions of how competitor suppression is applicable to our study via the rate-efficiency trade-off (LL 23-25 and 84-86),

Lines 110-120: Very clear and elegant argument.

Thank you.

Lines 131-132: I'm struggling to follow the logic here. Isn't it possible that the evolved producers enhanced sucrose, but not glucose, uptake?

Increasing sucrose uptake is a potential mechanism for producers to increase fitness in this environment. However, the competing non-producers can also uptake sucrose (via AGT1) and internally hydrolyse it with MAL12 using the same mechanisms as producers, so the suggested adaptation would not be exclusive to producers. Hence, we focussed on mechanisms that would be specific to producers, particularly because we observed that they increased relative fitness in competition with non-producers (Fig. 1a). We have added description of our motivation behind this approach (lines 119-122).

Lines 170-171: Again, I'm unclear on how it would suppress exploitation?

We have clarified our hypothesis that overproducers suppress competitors' benefits of exploitation by reducing the exploiters' metabolic efficiency (Lines 187-188). The mechanism behind this hypothesis is explained in more detail in the Mechanism 1 section of the results (lines 222-236), including a new schematic in Fig. 4e, a version of which was previously in Supplementary Fig. 8).

The key point here is that while both non-producers (NP) and producers (P) suffer a reduction in metabolic efficiency when in an environment where producers overexpress (P have a lower efficiency than NP due to their preferential product capture), the convex nature of the trade-off leads to a smaller difference in efficiency between NP and overproducers (either evolved or +SUC2) than between NP and wt. (Fig. 4e).

Line 217: The authors use carrying capacity to measure metabolic efficiency. More justification and explanation of this choice is needed.

We have now provided more detail. In particular, researchers often use growth yield (i.e. unit of growth (e.g. CFU) per unit of resource (e.g. mM of glucose) as a measure of metabolic efficiency, using biomass yield as a proxy for ATP yield (Ref: 60 = Pfeiffer et al 2001, Science). In the original version of the manuscript, our measure of yield was taken as carrying capacity (K), at a time by which sugars would be exhausted (as demonstrated for higher sugar concentrations than used here: doi: [10.4161/cc.8.8.8287](https://doi.org/10.4161/cc.8.8.8287)). This was justified because the strains being compared were grown in the same nutrient environments (i.e. 1% or 0.05% sucrose SC media). In the revised manuscript, we have scaled these measurements to $K/(\text{units of resource})$, which enables additional comparison between efficiencies in environments with different resource concentrations. Please see Results (line 238-239) and Methods (line 573-575)

Lines 220-222: I see why this reduced efficiency would exist, but shouldn't efficiency also be reduced for the producer with higher hexose concentrations?

Apologies for not making this clearer, please see our response to your comment regarding lines 170-171 (above), where we explain this point.

Lines 234-241: In this chunk of text, fitness should be taken to mean "fitness relative to non-producers", right? That is, saying +SUC2 is expected to have lower fitness than the wild type is meant to say + SUC2 is expected to have lower fitness compared to non-producers than the fitness of the wild type compared to non-producers. This point should be made explicitly, I struggled to follow the logic here at first, because I was not thinking of the appropriate measure/comparison for fitness.

Your understanding is correct. As suggested, we now clarify this explicitly (lines 252-253 and 256).

Lines 365-366: Thinking it over again, is it possible that the uptake of sucrose is upregulated as a threshold effect (i.e., more transporters are only produced given a sufficiently high sucrose concentration)? If this is the case, then perhaps +SUC2 loses its competitive advantage at low sucrose concentrations not because of an efficiency tradeoff, but rather because sucrose concentrations are not high enough to trigger the increase in uptake efficiency. Is there an argument that could rule out this alternative explanation?

Thank you for this interesting point. Based on the section of the manuscript being discussed, we assume that the reviewer meant to discuss hexose transporters rather than sucrose transporters. We have conducted additional experiments to rule out your suggested alternative explanation. Originally, we assumed that this hexose transporter upregulation would still apply at low sucrose concentrations (0.05%) because HXT2 is upregulated at very low hexose concentrations (i.e. 0.001% - S. Fig. 10 a-b) and +SUC2 has elevated growth rates in 0.05% sucrose (S. Fig. 8a).

To test this assumption, we conducted new experiments that examine HXT2 regulation in the wt and +SUC2 at 0.05% sucrose. In support of this assumption, we found that +SUC2 had higher expression than the wt over two experimental repeats (S. Fig. 10e-h). We have added this result to Mechanism 2 (lines 306-308) and included discussion to address this query (line 308-312 and 398-404).

Lines 389-395: The results from Scott (2022, <https://doi.org/10.1073/pnas.2214827119>) appear relevant to this paragraph.

We have now added the suggested reference (Scott 2022) and relevant discussion (line 462).

Lines 420-423: A very nice point!

Thank you. This is something that we are now working on.

Discussion: There have been a number of theoretical studies that consider how partial privatization can lead to the evolution of public goods production, but the authors only cite Gore et al (2009). It might be nice to try to make a few more connections between the results of this study and previously developed theory. See, for example, Inglis et al (2016, <https://doi.org/10.1098/rspb.2015.2682>), Estrela et al (2016, <https://doi.org/10.1111/1462-2920.13028>), and Lerch et al (2022, <https://doi.org/10.1371/journal.pcbi.1010666>).

Thank you for your suggestion. We have now added discussion of the work on partial privatisation (ll 419-425, including citing Lerch et al 2022) alongside other studies that we originally referenced (Gore et al 2009 and Bachmann et al 2010). We also discuss other references related to partial privatization (e.g., Lines 467-472: Neihus et al 2017; Jin et al 2018).

Line 553: Is there a reference to this definition of a selection coefficient? To me, this looks more like a relative fitness (fitness of one type divided by fitness of the other). In contrast,

selection coefficients (at least in the realm of population genetics) tend to have the form (relative fitness of one type) / (relative fitness of the other type) – 1 (see, for example, Otto and Day Chapter 3). I understand that it provides the relevant information, so this is only a semantic concern, but it is good to keep consistent terminology with past literature when possible.

Thank you for pointing out this potential misunderstanding. We have changed the terminology of our measure from “selection coefficient” to “normalised relative fitness”.

Finally, I'd like to make some suggestions that the authors may find useful but should not preclude publication. In many cases, these are recommendations for how to make the writing clearer (in my opinion).

Abstract: I found that the description of the study and results was somewhat brief given how much background was provided. For example, no information is provided about invertase. Further, neither “competitor suppression” nor “increasing product capture” are completely self-explanatory but these are key to the results. In contrast, lines 12-22 provide quite a bit of detail before mentioning the present study. The sentence on lines 14-16 is of course nice motivation but I wonder if it's needed in an abstract that feels rushed on the details of the study.

We have rewritten the Abstract to take the above suggestions into account.

Lines 77-78: The authors provide a useful summary of their study at the end of the introduction. This sentence is rather vague, however. It would be nice to tell the reader in a bit more detail why overproduction can evolve in this experiment even in competition with non-producers. As written, this chunk feels like a bit of a cliffhanger.

We have now added a description of the underlying mechanisms (Lines 82-86).

Lines 93-108: As someone that is not knowledgeable about specific mechanisms for microbial metabolism, this paragraph was challenging to understand. It would be clearer for readers that are uninformed about MAL, SUC2, etc., if the authors “flipped” this paragraph. That is, begin by explaining to the reader that in past studies the non-producer is dependent on the producer, which will not be the case in this study. With that background, the mechanistic/genetic details would be easier to follow.

We have now re-ordered this paragraph to improve clarity (as also suggested by reviewer #2).

Reviewer #2 (Remarks to the Author):

This study investigates the evolution of public goods production under conditions typically leading to production decline due to selection pressure by non-producing ‘cheats’. Using invertase production in *Saccharomyces cerevisiae* as a model trait/system, the authors find that – contrary to previous research – invertase over-producers evolve even in the presence of non-producers. Using a combination of different growth conditions and synthetic *S. cerevisiae* strains with modified secondary

features of sucrose metabolism, the authors identify two mechanisms, competitor suppression and increased product capture, which keep non-producers at bay and allow the evolution of overproducers.

In general, the work is well written, and the methods and results sections are robust and supported by the data presented in the figures. However, I have a general comment and a number of (mostly minor) concerns that prevent me from recommending publication in its current form.

My general concern relates to the approach of testing synthetic strains rather than evolved clones to identify mechanisms favoring the evolution of over-producers. In LL 383-388 of the discussion, the authors explain their reason for not following up their results with genome sequencing ('genetic basis of public-goods upregulation often unclear'). I partly sympathize with this view – but I still would have liked to see some screening of obvious candidates (such as the [promoters of] genes modified in the synthetic strains).

At the very least, I think that the authors should discuss potential differences between their engineered over-producer vs. the evolved clones (for instance regarding the extent of invertase up-regulation [not yet statistically compared?]) – and how these differences would change the importance of the two mechanisms favoring over-production.

Thank you for understanding our reasons for not deploying genetic sequencing in our study and thank you for an alternative suggestion which we have now implemented. As suggested, we have included a discussion of potential differences between the synthetic and evolved strains and how these differences would change the importance of the two mechanisms favouring over-production (LL 437-449). In particular, we have added a new plot with statistical comparisons of invertase activity between the engineered and evolved strains (Supplementary Fig. 4c) showing that our engineered strain had higher invertase activity. We subsequently present a rationale behind our assertion that this difference is not expected to qualitatively change the conclusions of our study (LL 439-456), also illustrated in the new Figure 4e.

Indeed, it might be worthwhile to consider introducing the general approach (i.e., testing synthetic strains instead of evolved clones) early on. I found it to be a strength of this study (and readers wouldn't go through the greater part of the manuscript expecting sequencing data).

Thank you for your suggestion and for acknowledging the strength of our approach. We have now emphasised our approach of deploying synthetic strains instead of evolved clones in the Abstract (LL 20-23), end of the Introduction (LL 78-81) and Discussion (LL 435-436).

Please find below a number of detailed comments that will hopefully help to further improve the manuscript.

Detailed comments:

Title: Consider removing the first part of the title ('Beating the cheat with kindness'). I think the second part is clear and straight to the point, and I find the mentioning of "kindness" somewhat misleading. After all, upregulation of public-goods production seems to evolve due to direct ("selfish") fitness benefits to producers, not due to indirect benefits (e.g., through altruism).

We have made the suggested edit.

LL 30-31: I suggest replacing the second 'products' by sth like 'metabolite' to make it clearer which products you are referring to in the following sentence.

We have now changed "products" to "metabolites".

L 39: Consider briefly explaining what 'prudently regulating expression' would look like.

We have added the following description: *by avoiding wasteful production by prudently regulating its expression.*

LL 46-56 (and LL 86-92, LL 345-347): The distinction between what you consider strong versus weak selection pressures from non-producers is not entirely clear to me. Do you consider strong selection pressures only to occur if non-producers can outcompete producers? In that case, I would suggest abandoning the strong/weak selection differentiation altogether, focusing instead on the different competitive outcomes (non-producer winning = strong selection [and narrow-sense cheating]; non-producer doing better, but still losing = weak selection).

We now clarify what we mean by strong/weak selection in our study. E.g. LL 46-48 now states "If under strong selection pressure from non-producers (*i.e. when non-producers can outcompete producers*), such as when spatial structure is low" and LL 50-52 now states "...when they are not exposed to strong selection pressure from non-producers (*i.e. when non-producers cannot outcompete producers*), such as in spatially structured environments...". The description has also been expanded in the re-written Results section (LL 91-115).

However, we felt it was important to keep "strong/weak" terminology in a number of instances in the manuscript while discussing past literature where it was not possible to determine the precise outcomes of competition (i.e. whether non-producers fully outcompeted or coexisted with producers).

Irrespective of this point, I also suggest that you make a clear distinction between (up)regulation (plastic response) and the evolution of upregulation (generally increased production). The references cited in L 48 contain examples of both (ref 17 vs 18).

As suggested, we have described the distinction between these different scenarios (LL 45-46) and added discussion (LL 453-455) to highlight differences between constitutive/facultative overexpression and how it can influence costs and competition outcomes.

LL 54-55 (& 78-80): This is generally true, but this study (<https://doi.org/10.1111/mec.16119>) reports that (a few) hyper-producers of pyoverdine evolved under conditions favoring cheating on this siderophore in *Pseudomonas aeruginosa*.

Thank you for pointing out this study, we have now provided a discussion of our results in the context of it (LL 421-425). We felt that Discussion section is a more suitable place to mention it (rather than the suggested lines 54-55 & 78-80 of the Introduction). This is because this study does not examine the relative fitness of the evolved overproducer, nor whether the overproduction is linked to the public goods nature of pyoverdine (i.e. it has diverse functions in, e.g., signalling and oxidative stress) meaning the overproduction might not be adaptive in terms of public good production.

L 73: Based on the pattern in Fig 1B, I would argue that they prevented fixation of non-producers, but not their invasion (frequency of producers declined over evolutionary times).

We have reworded this sentence accordingly.

L 78: 'prevent' instead of 'prevents'; missing 'from' after 'them'?

L 91: Consider adding a 'commonly used' or 'previously used' before 'S288c genetic background' for clarity.

L 93 (& L 97): 'locus' instead of 'loci'?

The suggested edits in these 3 comments have been made to the manuscript.

LL 93-108: I would find this paragraph easier to follow if you began by stating that the previously used S288c does not have an alternative way of metabolizing sucrose, and that you therefore constructed a suc2-deletion mutant in another background that had additional means to metabolize sucrose (MAL12 + AGT1), making it a stronger competitor (as is, the point of the paragraph only becomes apparent half-way through).

We have re-ordered this paragraph to address this point (and the similar comment made by reviewer #1).

Also, I suggest specifically stating here that you used a wild-type with the same background (not S288c) as producer.

We have now stated this in the re-ordered paragraph.

LL 105-108: What is a season, and how long does it last?

Details have been added here to specify a 24h batch culture season.

I'm also wondering about your differentiation between 'coexisting' (S288c producer and non-producer) and 'outcompeting' (CEN.PK2-1C producer and non-producer): Fig 1A shows that the CEN.PK2-1C non-producer outcompetes the producer, but it does not

show that the producer went extinct during one season. Unless this was the case, I would argue that they still co-existed over the duration/at the end of that season. Please clarify.

We have added description to the text to clarify our interpretation of the result (LL 111-114). We use single season, frequency-dependent competition experiments to predict long term dynamics. Namely, since producer relative fitness is below 0 across all initial frequencies in Fig. 1a, a well-established mathematical theory (which we now cite) indicates the competitive exclusion of producers in the long term.

This approach is frequently used in the field, including in papers cited in this article, (e.g. Ross Gillespie et al 2007 *AM Nat.* <https://doi.org/10.1086/519860>, Drescher et al 2014 *Current Biology* <https://doi.org/10.1016/j.cub.2013.10.030>, Lindsay et al 2019 *Nature Eco Evo.* <https://doi.org/10.1038/s41559-019-0944-9>).

Figure S2: Please describe the strains you used, e.g. in the figure legend (the reader will not have encountered some of the strains [+HXT7, +SUC2] when reading the manuscript and the supplements in parallel).

Strain descriptions have been added to S2 legend.

Figure 1: Please indicate in the legend which initial wt frequencies were tested (Fig.1A) – it is impossible to read out the exact frequencies for the four lowest levels from the graph. Please also indicate the number of replicates per frequency (they seem to vary between 3 and 4?). In Fig.1A, consider showing the regression line as a dashed/dotted line, which would make clearer that the relationship was not significant. Finally, the part of the legend referring to Fig.1B could be clearer regarding what's actually shown in the graph (e.g., 'b The frequency of producers initially declined, but then plateaued during a long-term evolution experiment). Finally, I would not consider an evolution experiment long-term, if it only runs for 10 cycles (also relevant for L 161).

We have edited Figure 1 and the legend according to these suggestions (and Supplementary Fig. 9 where the same applies re: dashed lines). We have also changed terminology from “long-term” to “multiple-season”.

LL 110-112: Please indicate the number of independent lines/replicates of your evolution experiment.

Added (LL 119 and 133),

LL 112-114: Please indicate initial relative frequency of the producer (from Fig 1B, I gather that it was 10%) and describe the decline (e.g., ‘from x to y % of the population’) so that readers don't have to jump to the figure/methods.

This has been added (L123).

Along the same lines, please also explain here why you used this (relatively low) initial frequency (and why you used a much higher frequency in the experiments described in LL 160-162 and depicted in Fig 3B?).

The different initial frequencies were specifically chosen as part of different experimental designs. During the initial evolutionary experiments, populations were initiated with a minority of producers (approx. 10%) so that producers did not have a higher probability than non-producers of acquiring adaptive mutations in non-social traits by being numerically dominant in the population, as found previously (e.g. Asfahl et al 2015; Waite & Shou, 2012; Morgan et al 2012). On the other hand, higher initial frequencies in Fig 3b were used to test population resistance against invasion by non-producers, where invasion from rare is tested to give non-producers a relative fitness advantage due to negative-frequency dependent selection. We have now included some of the description from the methods into the Results section to highlight these different experimental design motivations (LL 119-122 and LL177-178).

LL 121-123: Is that one clone per line? And how many control lines did you run? Also, three?

This information has now been included (at the specified place (L 133), and in the section above in response to comment LL 110-112).

Figure 2: Looking at Fig. 2A, I'm wondering how the evolved producers did when compared to the ancestral non-producer instead of the wt? Did they have a higher fitness, and was this dependent on whether they had evolved with a non-producer vs. alone?

We have now added a new plot and associated analysis to show the relative fitness of these strains when in pairwise competition against the ancestral non-producer (Supplementary Fig. 3a).

L 144: Please briefly indicate here how invertase activity was measured (I'm not suggesting to provide any details, mentioning the nature of the assay is enough).

Brief description added.

LL 153-154: Fig.3A seems to suggest that +SUC2 actually wins in direct competition against the non-producer. Consider indicating that here.

While +SUC2 tended towards outcompeting the non-producer here, our statistical analysis indicates that +SUC2 was not significantly fitter than the non-producer ($p > 0.05$). We have added this new analysis to the Figure Legend (Fig. 3a).

Figure S4: Typo in second line (activity)

Corrected.

Figure 3: Please describe the strains in the figure legend. The reader will not yet have encountered +HXT7 when Figure 3 is first referenced.

We have added a brief description.

Figure 4: You describe how i) and iii) were tested, but what about ii)? Consider describing that here, too. Please also consider changing your numbering of the different parts of the graphs; using i), ii), etc. both for numbering the hypothesis and referencing specific parts of the figure is confusing.

Numbering format has been changed. We have added brief description of testing ii) (now “2”) to the legend.

LL 187-188: This sentence is hard to understand, consider rephrasing.

This has been rephrased.

LL 193-197: To me, the result that ‘agt1-deletion strains did not prevent the invertase overproducer (+SUC2 Δ agt1) having a selective advantage over its equivalent wt producer (Δ agt1)’ shows that the enhanced competitiveness of the SUC2 overproducers is not ONLY caused by increasing internal sucrose metabolism. But I don’t understand how you can exclude the possibility that it partly explains the increased competitiveness. To me, this would require directly comparing the magnitude of the respective competitive advantages. Please clarify.

As suggested, we had conducted the statistical test (Supplementary table 2) to show that Δ agt1+SUC2 has a greater competitive advantage than +SUC2. We have now made this result more overt in the main text (end of “Why does public-good overproduction promote producers?” section).

LL 264-265: Consider briefly giving the gist of the approach.

We have added a brief description.

L 281: I suggest replacing ‘this increase’ with ‘this qualitative difference’ (in HTX2 expression of +SUC2 vs. wt) for clarity.

We have made the suggested edit.

L 336: Consider adding an ‘only’ after ‘overproduction’ for clarity.

The results in Figueiredo et al 2021 (<https://doi.org/10.1111/mec.16119>) where overproducers did evolve in unstructured environments (as mentioned in the comment above: “LL 54-55 (& 78-80)”) means that it would not be appropriate to add “only”. Therefore, to improve clarity, we have more accurately re-phrased the sentence to “Consistent with this, the evolution of overproduction has **generally been observed** in spatially structured environments”.

LL 343-344: Also when populations grow/evolve in spatially-structured environments?

Yes, in these examples overproducers transiently evolved in spatially structured environments, but are later outcompeted by non-producers. We have added that this occurred in spatially structured environments (LL370-372).

L 373: Replace ‘producers’ with ‘over-producers’ (the wt is also a producer).

Done.

L 389-390: Do you mean it is the first that shows benefits of overproduction in a situation where non-producers can outcompete the wt?

We have now clarified what we meant here in terms of how overproduction can provide benefits to overproducers against exploitative competitors. We use the term exploitative competitors (rather than non-producers) because our findings also apply to scenarios where overproducers compete with comparatively lower producers, such as the wt (as discussed LL 168-174 and LL 381-384), which can, in this context act as “cheats” (Ghoul et al 2014). Please also see response to reviewer #3 regarding competitions against wt producers and new data S.Fig 5b.

L 404: ‘it’ = ‘invertase production’?

This has been specified.

L 410: I suggest adding ‘invertase’ before ‘non-producers’ – non-producers of other ‘public goods’ such as siderophores are not necessarily rare (e.g., <https://doi.org/10.1038/s41467-017-00509-4>)

Added “invertase” as suggested.

L 492-493: Did you inoculate different wells/replicates of the same treatment with different precultures (such that, say, different replicates of +SUC2 monoculture growth were started from different precultures)?

The description of the inoculation procedure has now been expanded. (LL 560-562).

Reviewer #3 (Remarks to the Author):

This paper is a strong investigation into the effects of invertase overexpression in *S. cerevisiae*. It is a tour-de-force that dissects exactly how overexpression of SUC2 can increase fitness, and it beautifully shows that it is through increasing the amount of glucose available in the external environment (which has the effect of decreasing metabolic efficiency of competitor cells) and through increasing the amount of hexose transporters for private glucose utilization. I have very little to contribute to the experimental work, save a minor comment (see below).

Thank you

The main issue I have with this paper is the framing of the results. It seems that overexpression of invertase is the result of selection due to sucrose in the environment and not the result of the presence of non-producers. Figure 2 clearly shows that strains increase invertase expression when evolved in the presence of sucrose without non-

producers. Thus, while the rest of the experimental results show the effect that increased invertase expression can have on non-producers (i.e., decreased metabolic efficiency at certain sucrose concentrations), that is not why it is selected/favored in the first place. These effects are by-products of selection for utilizing sucrose. Lines 26-28 in the Abstract are not quite accurate: “A combination of competitor suppression and concurrently increasing product capture were revealed to be driving the evolution of overproducers, in environments originally thought not to support public goods production.” Competitor suppression is not driving the evolution of overproducers because it occurs without non-producing competitors. The most likely case is the direct, private benefits that the cells receive that are driving the increase in invertase production. Since this particular trait has such a strong private component, it makes the results less about the evolution of public goods expression. I think the authors should revisit the framing of the paper.

We have re-worded our arguments to further emphasise that overproduction is favoured both in the absence or presence of non-producers, as well as adding new experimental data showing that engineered overproducers (+SUC2) outcompete the wt in direct competition (Supplementary Fig. 5b). We argue that this competitive advantage is based on the same mechanisms of competitor suppression and increased product capture. Namely, in the absence of non-producers, the overproducers compete against the wt-producers with comparatively lower invertase production levels, that in this context act as “cheats” (Ghoul et al 2014) (see new discussion LL 168-174 and LL 381-384). Thus, the same arguments regarding competitor suppression (Fig. 4e) and increased product capture (Fig. 7a) would hold as in the case when overproducers compete with non-producers, who are also considered as “cheats”.. We have also re-worded the Abstract (LL 22-26) and the manuscript to reflect this. While we appreciate that there is a strong private component to the maintenance of SUC2 production, we feel that it is still important to frame it in the context of public goods expression, both to be consistent with existing literature (e.g. Celiker & Gore 2012; MacLean & Brandon 2008; Sanchez & Gore 2013; Koschwanez et al 2013; Lindsay et al 2019), and also because the private and public elements are intricately linked, as we show in this study.

Minor comments:

Figure 3b: Why is there such a big difference in the initial proportion of producers in the transfer experiments in 1b and 3b?

This point was also raised by Reviewer #2, and the same response applies. In particular, the different initial frequencies were specifically chosen as part of different experimental designs. During the initial evolutionary experiments, populations were initiated with a minority of producers (approx. 10%) so that producers did not have a higher probability than non-producers of acquiring adaptive mutations in non-social traits by being numerically dominant in the population, as found previously (e.g. Asfahl et al 2015; Waite & Shou, 2012; Morgan et al 2012). On the other hand, higher initial frequencies in Fig 3b were used to test population resistance against invasion by non-producers, where

invasion from rare is tested to give non-producers a relative fitness advantage due to negative-frequency dependent selection. We have now included some of the description from the methods into the Results section to highlight these different experimental design motivations (LL 119-122 and LL177-178).

Also, in Figure 1b, the proportion of wt producers seems to level-off, which the authors suggest is solely due to adaptation on the part of the producers. Why don't the wt producers also reach a plateau in Figure 3b (it is the same amount of transfers). Is it possible that the results in Figure 1b would be better interpreted as a combination of an increase in fitness, as well as frequency dependent selection?

This is an interesting point. We argue that the producers do not reach a plateau in Fig. 3b (unlike 1b) because the populations are initiated from very different initial frequencies, even if the same adaptation had hypothetically occurred. Therefore, by season 10, the evolved producers in 3b would not be benefitting from frequency dependent selection (FDS) to the same extent as in 1b because they are not rare in the population.

Regarding 1b resulting from an increase in fitness **and** FDS, in agreement, we suggest that this result arises due to overproducer evolution, which subsequently gives rise to FDS between the evolved strains (because they level off in 1b (low producer frequency) but not 3b (high producer frequency)). However, we highlight that overproducer evolution is the important cause of 1b levelling off because FDS was not detected between the ancestral strains (Fig. 1a), even at very low initial producer frequencies (~1%) which is lower than those of 1b. Nevertheless, we do discuss FDS in our study and cite relevant past studies (e.g. Gore et al 2009; Ross-Gillespie et al 2007).

Typos: Line 93 Not sure if the authors would like this to be singular or plural. I think plural, so remove "an".

This section has been corrected during the edits suggested by reviewers #1 & #2.

Line 152: When you say "on" a certain medium, it usually indicates agar-based solid medium. Better to replace with "in" for liquid.

We have made the suggested edit, (also to Supplementary Fig 5).

Reviewer #4 (Remarks to the Author):

The authors of the study used the yeast *Saccharomyces cerevisiae* as a model system to challenge the conventional understanding of microbial production of public goods. Contrary to past studies, they found that overproducers can evolve even under strong selection pressure from non-producers. This study claims to have identified novel mechanisms, including competitor suppression and increased product capture, as driving the evolution of overproducers.

I have read other papers from the group and found the results appealing. However, unlike the other papers, which were easy and pleasant to read, this manuscript was very difficult to follow, from the abstract to the main text.

- The general writing lacks a logical flow that is capable of guiding the reader through the main findings.

We have edited the manuscript to clarify the message (please see responses to other reviewer comments for specific edits).

- Abstract needs to be restructured; it has a very long introduction that doesn't seem to be necessary while also doesn't communicate the relevance of the work.

As suggested, we have fully rewritten the Abstract so less of it covers the background and more is given to explain the details, findings and relevance of the current study.

- The introduction seems to be repetitive (first two paragraphs) and doesn't present a clear narrative.

We have made edits to the introduction and manuscript to make the narrative of the paper clearer and remove repetitions. We hope that it is now clear that the first two paragraphs make distinct points: Paragraph 1: Introduces the importance of microbial secreted products (public goods), the threat of exploitative competitors, and general mechanisms that have been identified to prevent producer exclusion. Paragraph 2: Describes the current understanding of how microbes adjust the levels of public good production in response to different selection pressures and sets the stage for the tested hypothesis.

- The figures and captions should provide a clear take-away.

We have added titles/descriptions to the Figure Legends to aid with interpretation and take-away message.

Unfortunately, I do not recommend the publication of this manuscript in its current form. However, I encourage the authors to restructure the findings (text and figures) to tell a better story that may be interesting to a broader audience.

** See Nature Portfolio's author and referees' website at www.nature.com/authors for information about policies, services and author benefits.

Reviewers' Comments:

Reviewer #2:

Remarks to the Author:

I have reviewed the initially submitted version of this manuscript. In this revised version, the authors have addressed all my comments to my satisfaction. I only have a small number of additional (minor) comments that the authors can easily address while finishing their manuscript for publication. Congratulations on an interesting and thought-provoking study!

Detailed comments:

L 37: It is grammatically unclear to what „its” refers.

LL 55-56: I suggest deleting this sentence (“However, it can be costly...”). To me, it is unclear without further explanation (does that mean that the evolution of higher production levels is ultimately always transient?) and does not add to the point. Maybe you could briefly explain this point below, in the discussion (LL 370-372)?

LL 204-205: Growth was impeded when compared against what?

LL 211-212: Maybe add an “intact” before “internal sucrose metabolism”? I was confused because this wording makes it sound as if +SUC2 has only the internal sucrose metabolism. I was also wondering why +SUC2 Δ agt1 has a higher relative fitness than +SUC2? Because AGT-dependent sucrose import is so energetically costly under the experimental conditions?

L 214: Consider replacing “c.f” by “versus”.

LL 263-266: This is of course true. But isn't the action of a second mechanism already suggested by the fact that this mechanism can only reduce, but not reverse, the costs of exploitation by non-producers?

L 375: “It” instead of “them” (refers to overproduction)?

LL 476-479: This sounds as if the changes are due to phenotypic plasticity rather than genetic change. Please rephrase.

Reviewer #3:

Remarks to the Author:

After a careful reading of the revised manuscript and associated files, I believe the authors have addressed all my concerns.

Reviewer #4:

Remarks to the Author:

The authors have addressed my previous concerns and made significant improvements to the paper. The revisions have enhanced the clarity and overall quality of the manuscript.

Response to Reviewers (responses in blue)

Reviewer #2

I have reviewed the initially submitted version of this manuscript. In this revised version, the authors have addressed all my comments to my satisfaction. I only have a small number of additional (minor) comments that the authors can easily address while finishing their manuscript for publication. Congratulations on an interesting and thought-provoking study!

Detailed comments:

L 37: It is grammatically unclear to what „its” refers.

We have replaced “its expression” with “the expression of public goods”.

LL 55-56: I suggest deleting this sentence (“However, it can be costly...”). To me, it is unclear without further explanation (does that mean that the evolution of higher production levels is ultimately always transient?) and does not add to the point. Maybe you could briefly explain this point below, in the discussion (LL 370-372)?

Minor edits and additions have been made to the suggested sections to improve clarity.

LL 204-205: Growth was impeded when compared against what?

We have edited this sentence to clarify that this was meant in absolute terms.

LL 211-212: Maybe add an “intact” before “internal sucrose metabolism”? I was confused because this wording makes it sound as if +SUC2 has only the internal sucrose metabolism.

We have added “intact” as suggested.

I was also wondering why +SUC2 Δ agt1 has a higher relative fitness than +SUC2? Because AGT-dependent sucrose import is so energetically costly under the experimental conditions?

Note that the relative fitness values being compared are normalised against equivalent genotypes with wt SUC2 production levels. Thus, +SUC2 Δ agt1 is being compared against Δ agt1, while +SUC2 is being compared against wt (rather than +SUC2 being directly competed against +SUC2 Δ agt1). For clarity, we have now added “**normalised** relative fitness” (Line 212 (new version: Line 207). The reasons for +SUC2 Δ agt1 being higher than +SUC2 are discussed in Lines 214-218. Briefly, rather than the energetic costs of AGT1-dependent sucrose import (as suggested), it relates more to the fact that the wt loses more relative fitness (and growth rate) than +SUC2 when AGT1 is deleted (Supplementary Fig 7a).

L 214: Consider replacing “c.f” by “versus”.

Replaced as suggested.

LL 263-266: This is of course true. But isn't the action of a second mechanism already

suggested by the fact that this mechanism can only reduce, but not reverse, the costs of exploitation by non-producers?

Yes, you are correct, and this is described in Lines 257-263 (New version: 253-257). The intuition is then summarised in Lines 263-266 (New version: 259-262). For clarity, we have added “in summary” (Line 264, new version: 259).

L 375: “It” instead of “them” (refers to overproduction)?

Changed as suggested.

LL 476-479: This sounds as if the changes are due to phenotypic plasticity rather than genetic change. Please rephrase.

We have rephrased to improve clarity. In particular, we removed the use of the term “physiological plasticity”.